

**Parameterization of the light absorption properties of chromophoric dissolved organic**
**matter in the Baltic Sea and Pomeranian Lakes**
Justyna Meler[a*], Piotr Kowalczuk[a], Mirosława Ostrowska[a], Dariusz Ficek[b], Monika
Zabłocka[a], Agnieszka Zdun[a]
[a] Institute of Oceanology Polish Academy of Sciences, Powstańców Warszawy 55, 81-712
Sopot, Poland
[b] Institute of Physics, Pomeranian University of Słupsk, Bohaterów Westerplatte 64, 76-200
Słupsk, Poland
[*] corresponding author: jmeler@iopan.pl
Keywords: Baltic Sea; Pomeranian lakes; Chromophoric Dissolved Organic Matter; three
alternative models of CDOM absorption; light absorption; ocean optics
**Abstract**
This study presents three alternative models for estimation of absorption properties of
Chromophoric Dissolved Organic Matter, $a_{\mathrm{CDOM}}(\lambda)$. For this analysis we used a database
containing 556 absorption spectra measured in 2006 – 2009 in different regions of the Baltic
Sea (open and coastal waters, the Gulf of Gdańsk and the Pomeranian Bay), at river mouths,
in the Szczecin Lagoon and also in three Pomeranian lakes in Poland – Lakes Obłęskie,
Łebsko and Chotkowskie. Observed variability range of the CDOM absorption coefficient at
400 nm, $a_{CDOM}(400)$, contained within $0.15 - 8.85$ m$^{-1}$. The variability in $a_{CDOM}(\lambda)$ was
parameterized with respect to three orders of magnitude variability in the chlorophyll *a*
concentration *Chla* ($0.7 - 119$ mg m$^{-3}$). Chlorophyll *a* concentration and CDOM absorption
coefficient, $a_{\mathrm{CDOM}}(400)$ were correlated, and statistically significant, non-linear empirical
relationship between those parameters was derived ($R^2$=0.83). Based on observed co-variance
between these parameters, we derived two empirical mathematical models that enabled to
project the CDOM absorption coefficient dynamics in natural waters and reconstruct the
completed CDOM absorption spectrum in the UV and visible spectral domains. The first
model used the chlorophyll *a* concentration as the input variable. The second model used the
$a_{\mathrm{CDOM}}(400)$, as the input variable. Both models were fitted to power function and the second
order polynomial function was used as the exponent. Regression coefficients for derived
formulas were determined for wavelengths from 240 to 700 nm at 5 nm intervals . Both



approximation reflected the real shape of the absorption spectra with low uncertainty. Comparison of these approximation with other models of light absorption by CDOM proved that proposed parameterizations were better (bias from -1.45% to 62%, RSME from 22% to 220%) for estimation CDOM absorption in optically complex waters of the Baltic Sea and lakes.

## 1. Introduction

All natural waters contain optically significant constituents that determines their inherent optical properties: absorption coefficient, scattering coefficient and beam attenuation coefficient. The total absorption coefficient in ultraviolet and visible spectral range of the electromagnetic radiation spectrum, is almost entirely determined by four main groups of absorbents: water molecules, organic and inorganic suspended particulate matter (SPM), and Chromophoric Dissolved Organic Matter (CDOM). The quantity and qualitative properties of these absorbents significantly affect the quantity and spectral distribution of light in the aquatic environment. The absorption of pure water measured by Pope and Fry (1997) is almost constant in natural waters and may be omitted in further analysis because it does not contribute to a variability of total absorption coefficient. Changes in spectral values of the pure sea water are almost entirely determined by the concentration and the composition of sea salt ions and dissolved gases, and is mostly pronounced in the UV-A and UV-B spectral region below 300 nm (Woźniak and Dera, 2007). Spectral properties (values and spectral shape) and the mutual proportions of light absorption coefficients by CDOM, $a_{CDOM}(\lambda)$, by phytoplankton pigments, $a_{ph}(\lambda)$, organic detritus and mineral particles $a_{NAP}(\lambda)$, determine the spectral shape and magnitude of the total absorption spectrum as well as affects both the inherent and the apparent optical properties of natural waters (Woźniak and Dera, 2007).

The Chromophoric Dissolved Organic Matter is the uncharacterized fraction of the dissolved organic matter pool consisting from heterogeneous mixture of water soluble organic compounds that have ability to absorb light (Nelson and Siegel, 2002). The effect of the CDOM absorption is mostly visible in the UV and blue spectral range of electromagnetic radiation, where CDOM contribution to the total non-water absorption could reach 90%, even in the clearest natural waters found in South Pacific Subtropical Gyre south off Easter Island, (Morel et al., 2007; Bricaud et al., 2010; Tedetti et al., 2010). Presence of high concentration of CDOM usually shift the spectral maximum of the water transparency to solar radiation and water leaving radiance toward the longer wavelength (Darecki et al., 2003; Morel and Gentili,





2009). In extreme cases, in humic boreal lakes, the CDOM reduces the water leaving radiance

intensity in the visible spectrum almost to null (Ficek et al., 2011; Ficek et al., 2012; Ylöstalo

et al., 2014). CDOM absorption band overlaps also with primary phytoplankton pigment

absorption band in the blue part of the spectrum contributing to significant errors of standard

algorithms for retrievals of chlorophyll *a*, especially in costal ocean and shelf and semi-

enclosed seas (Darecki and Stramski, 2004; Siegel et al., 2005). Therefore, appropriate

quantitative and qualitative descriptions of the optical properties of CDOM is crucial in the

ocean color remote sensing of aquatic environments.

CDOM plays also various ecological roles in aquatic environments: even small

concentrations strongly absorb UV radiation, protecting organisms from its destructive action.

Higher levels of CDOM absorptions limits the amount of radiation available for

photosynthesis and consequently reducing the primary production of organic matter in that

water (Górniak, 1996; Wetzel, 2001). CDOM plays an important part in various biological

processes taking place in water bodies: it can affect the species composition, number and size

of plankton organisms (Arrigo and Brown, 1996; Campanelli et al., 2009), and in oligotrophic

lakes can promote the growth of bacterioplankton (Moran and Hodson, 1994). Several authors

have pointed out that CDOM is a potential source of reactive oxygen forms in aquatic

ecosystems, which has a considerable influence on a variety of biological processes

(Whitehead and de Mora, 2000; Kieber et al., 2003).

CDOM absorption decreases exponentially towards longer wavelengths and can be

described by the exponential function (Jerlov, 1976, Bricaud et al., 1981, Kirk 1994):

$$a_{CDOM}(\lambda) = a_{CDOM}(\lambda_0)e^{-S(\lambda_0 - \lambda)} \tag{1}$$

where: $a_{CDOM}(\lambda)$ is the light absorption coefficient for a given wavelength $\lambda$, $\lambda_0$ is the

reference wavelength, and $S$ is the slope of the spectrum within a given wavelength interval.

The CDOM accumulates in the surface Baltic Sea waters as a combined effect very

high inflow of fresh water from rivers and the limited exchange of waters with the North Sea

and very high productivity of this marine basin, (Kowalczuk et al., 2006). The systematic

studies over the last two decades on optical properties in the Baltic Sea waters and adjacent

fresh water systems coastal lagoons and Pomeranian lakes, provided evidence that the CDOM

is the principal absorbent of solar radiation and the main factor governing their optical

properties (Kowalczuk 1999; Kowalczuk et al., 2005; 2006; 2010; Ficek et al., 2012; Ficek

2013).



The main objective of the present work was to derive three alternative
parameterizations scenarios of the relationships between the CDOM absorption coefficient in
the Baltic and Pomeranian lakes waters and physical or biogeochemical variables. We have
performed analyses using combined data set of optical properties of marine and lacustrine
water samples, treating the data as a single, pooled set. Optical properties of lacustrine waters
displayed a resemblance to marine waters in the Baltic Sea, despite observed differences in
trophic status of those water bodies. According to Choiński (2007), lakes waters were divided
into ultraoligotrophic, oligotrophic, mesotrophic, eutrophic, hiperetrophic and dystrophic. The
trophicity is determines by the concentration of chlorophyll $a$, water transparency determined
by Secchi disk, and the concentration of biogenic factors, e.g. nitrogen and phosphorus
(Carlson, 1977; Kratzer and Brezonik, 1981). The ranges of concentrations of chlorophyll and
nutrients defining trophicity are higher than in marine waters. In our modelling approach we
have assumed that lakes could be treated as a natural extension of coastal, lagoon and river
mouth waters. The motivation for development of these models was to estimate a complete
spectrum of the CDOM light absorption coefficients by using different input parameters: $i$) in
the first scenario the known chlorophyll $a$ concentration, $ii$) in the second scenario known
value of the CDOM absorption coefficient at 400 nm, $a_{CDOM}(400)$, $iii$) and in the third
scenario known value of $a_{CDOM}(400)$ and known nonlinear relationship between CDOM
absorption coefficient and the spectral slope coefficient $S$. Developed models can be used to
improve the accuracy of ocean colour remotes sensing algorithms for retrieval of
environmental variables in the Baltic Sea, adjacent estuaries and lagoons and fresh water
lakes.
**2. Material and methods**
2.1 *Sampling area*
Water samples for determination of optically significant water constituents
concentrations were collected from August 2006 to November 2009 in the southern Baltic and
in three lakes in the Pomeranian Lake District (Poland) during the long term observation
program of inherent and apparent optical properties for calibration and validation of ocean
colour satellite imagery products conducted by the Institute of Oceanology, Polish Academy
of Sciences, Sopot, Poland, (IOPAN). Location of 116 measuring stations, where empirical
data were gathered (a total of 413 data sets) during 16 cruises of r/v Oceania on the Baltic
were shown on Figure 1, and cruises details is given in the Table 1. Research cruises were



organized to capture the dynamics of natural seasonal variability occurring in temperate
waters: *i*) at the end of the winter before the onset of the spring phytoplankton bloom, when
wind-driven mixing, the vertical convective thermohaline circulation, reduced biological
activity and reduced riverine outflow all result in clearer surface waters; *ii*) in spring when the
spring phytoplankton bloom coincides with maximum freshwater runoff from Baltic Sea
watershed; *iii*) and at the end of summer at the peak of secondary phytoplankton blooms and
the period of maximal thermal stratification of waters. The geographical coverage of the
samples included the Gulf of Gdańsk, the Pomeranian Bay, the Szczecin Lagoon, Polish
coastal waters and the open sea (the Baltic Proper). The coastal sites in the Gulf of Gdańsk
and the Pomeranian Bay are under the direct influence of two major river systems, the Vistula
and the Odra, which drain the majority of Poland. Additionally samples were collected twice
a month on sampling station at the Sopot pier (Gulf of Gdańsk), from which 66 sets of data
were obtained. Field observation were also carried from April 2006 to November 2009 on
monthly intervals a month (except months when the surface of the lake was covered with ice)
in three Pomeranian lakes (Łebsko, Chotkowskie and Obłęskie) from which 77 data sets were
obtained. Selected lakes are closed water bodies with only small rivers flowing in and out of
them. Lake Łebsko is a specific case: it is a coastal lake and connected directly to the sea by a
short canal. Part of Lake Łebsko area immediately adjacent to the canal can, on occasion, be
inundated when large backflows of sea water enter the lake. The lake's water level can then
rise by 50-60 cm (Chlost and Cieśliński, 2005). Such a situation obviously affects the
composition and properties of the lacustrine water. Similar effects, resulting from the great
variability of water properties, can be expected at the points where rivers flow into lakes. The
lacustrine water in these areas is thus modified by the river water.
2.2 *Samples processing*
Discrete samples of water were taken from the surface layer of the southern Baltic and
the three Pomeranian lakes with use of the Niskin bottle. The samples for spectroscopic
measurements CDOM light absorption underwent a two-step filtration process. The first
filtration was through acid-washed Whatman glass fibre filters (GF/F, nominal pore size 0.7
µm). The water was then passed through acid washed Sartorius 0.2 µm pore cellulose
membrane filters to remove fine-sized particles. Spectrophotometric scans of CDOM
absorption spectra were performed with use the Unicam UV4-100 double beam
spectrophotometer installed both in land base laboratory and on board of the research ship in
the 240-700 nm spectral range. The cuvette pathlength was 5 cm and the MilliQ water was





used as the reference for all measurements. The absorption coefficient $a_{CDOM}(\lambda)$ was
calculated using the following equation:
$$a_{CDOM}(\lambda) = 2.303 \cdot A(\lambda)/L, \qquad (2)$$
where: $A(\lambda)$, is the optical density, L is the optical path length in meters and the factor 2.303
is the natural logarithm of 10.
A nonlinear least squares fitting method using a Trust-Region algorithm implemented
in Matlab R2009 was applied (Stedmon et al., 2000, Kowalczuk et al., 2006) to calculate
CDOM absorption spectrum slope coefficient, S, in the spectral range 300-600 nm using the
following equation:
$$a_{CDOM}(\lambda) = a_{CDOM}(\lambda_0)e^{-S(\lambda_0 - \lambda)} + K \qquad (3)$$
where: $\lambda_0$ is 350 nm, and K is a background constant that allows for any baseline shift caused
by residual scattering by fine size particle fractions, micro-air bubbles or colloidal material
present in the sample, refractive index differences between sample and the reference, or
attenuation not due to CDOM. The parameters $a_{CDOM}(350)$, S, and K were estimated
simultaneously via non-linear regression using Equation 3.
The chlorophyll $a$ concentration was determined with use pigment extraction method.
Pigments contained within suspended particles were collected by filtration of water samples
onto 47-mm Whatman glass-fiber filters (GF/F) under low vacuum and extracted 24 hours in
96% ethanol at room temperature. Chlorophyll $a$, $Chla$, concentration was determined
spectrophotometrically with a UV4-100 spectrophotometer (Unicam, Ltd). In this method the
optical density (absorbance) of pigment extract in ethanol was measured at 665 nm. After
correction for background signal in the near infrared (750 nm): $\Delta OD = OD(665nm)$ -
$OD(750nm)$, the absorbance was converted to chlorophyll $a$ concentration, using an equation
involving the volumes of filtered water ($V_w$) [dm$^3$], and ethanol extract ($V_{EtOH}$) [cm$^3$], a 2-cm
path length of cuvette (l), and the chlorophyll $a$ specific absorption coefficient in 96% ethanol
[dm$^3$ (g cm)$^{-1}$] [Stricland and Parsons 1972; Stramska et al., 2003]:
$$Chla = (10^3 \cdot \Delta OD \cdot V_{EtOH})/(83 \cdot V_w \cdot l)^{-1}. \qquad (4)$$
During field surveys temperature and salinity profiles were measured with and
SeaBird SB36 CTD probe to provide background physical conditions during sampling.





The collected data were analyzed by the use of statistical package and data
visualization software (SigmaPlot 8.1). Dynamic range of variability of analyzed optical
parameters values exceeded 3 orders of magnitude, therefore logarithmic transformation was
applied which allowed better presentation of their dynamics changes and to analyze
statistically collected data set accordingly. Following arithmetic and logarithmic statistical
metrics were used to assess uncertainty of developed empirical relationships and models:
• relative mean error (systematic): $\langle \varepsilon \rangle = N^{-1} \sum_i \varepsilon_i$ (where $\varepsilon_i = (X_{i,C} - X_{i,M})/X_{i,M}$ ); (5a)
• standard deviation (statistical error) of $\varepsilon$ (RMSE – root mean square error):
$$\sigma_\varepsilon = \sqrt{\frac{1}{N}\left(\sum (\varepsilon_i - <\varepsilon>)^2\right)}$$    (5b)
• mean logarithmic error: $\langle \varepsilon \rangle_g = 10^{\left[\langle \log(X_{i,C}/X_{i,M})\rangle\right]} - 1$    (6)
• standard error factor: $x = 10^{\sigma_{\log}}$    (7)
• statistical logarithmic errors: $\sigma_+ = x - 1 \quad \sigma_- = \frac{1}{x} - 1$    (8)
where $X_{i,M}$ - measured values; $X_{i,C}$ - estimated values (subscript $M$ stands for 'measured';
subscript $C$ stands for 'calculated');
• $\langle \log(X_{i,C}/X_{i,M})\rangle$ - mean of $\log(X_{i,C}/X_{i,M})$;
• $\sigma_{\log}$ - standard deviation of the set $\log(X_{i,C}/X_{i,M})$.
**3. Results**
3.1 *Variability of analysed parameters and empirical relationship between CDOM absorption*
*and spectral slope coefficient.*
Variability range and average values if selected optical parameters: the light absorption
coefficients by CDOM at two wavelengths: 375 and 400 nm; $a_{CDOM}(375)$ and $a_{CDOM}(400)$;
spectral slope $S$, and chlorophyll $a$ concentrations, *Chla,* measured in the study area and used
for formulation of empirical model have been presented in the Table 2. The minima in of the
variability ranges of $a_{CDOM}(375)$, $a_{CDOM}(400)$ and *Chla,* were noted in marine waters. The
minimal values of CDOM absorption coefficients in lacustrine waters were almost an order of





magnitude higher than in marine waters, indicating significant accumulation of CDOM in
fresh waters. The maximal values of $a_{CDOM}(375)$, $a_{CDOM}(400)$ and *Chla* were observed in
fresh waters, maximal values were approximately two time higher than values of respective
parameters in marine waters. Consequently average values of the CDOM absorption
coefficients: $a_{CDOM}(375)$, $a_{CDOM}(400)$, chlorophyll *a* concentration were higher in fresh waters
compared to marine waters. The reverse trend is observed CDOM absorption spectrum slope
coefficient, *S*, variability range: both of minimal and maximal spectral slope values were
lower in the lakes than those observed in the marine waters. The average value of the spectral
slope coefficient was higher in marine waters than in lake waters. These two data sets,
measured in the Baltic waters and Pomeranian lakes were statistically significantly different,
as indicated by results of simple analysis of variance: ($p = 3.4 \cdot 10^{-38}$). However, their
variability ranges were such that the data from the two different aquatic environments were
overlapping creating coherent data set, that could be analysed together. Our principle
assumption for the derivation of CDOM absorption model was that, the optical properties of
lacustrine waters could be treated as they were an extension of estuarine and marine waters.

The spectral slope coefficient was inversely non-linearly related with the CDOM

absorption coefficient. The highly absorbing samples were spectrally flatter (characterised by
lower *S* value). Different functional types were used to model this relationships: hyperbolic
(Stedmon and Markager, 2001, Kowalczuk et al., 2006), or logarithmic (Kowalczuk et al.,
2005). For consistency with Kowalczuk (2001) we have used the log–linear fit to describe the
relationship between $a_{CDOM}(400)$ and *S*. The distribution of the spectral slope in the function
of CDOM absorption coefficient in the Baltic Sea (black dots) and Pomeranian lakes (green
dots) has been presented on the Figure 2a. The black line presents log-linear dependence
(Equation 9),obtained by Kowalczuk (2001), overlaid on our data set:
$$S = log[1.038\ a_{CDOM}(400)^{-0.022}]. \qquad (9)$$

The old realtiohsip worked satisfactory for part of Baltic Sea data set ($R^2 = 0.76$), but

it does not cover large group of CDOM absorption coefficients values larger than 5 m$^{-1}$. The
$a_{CDOM}(400) > 5$ m$^{-1}$ were measured in the lakes and in estuarine waters and in the Szczecin
Lagoon and Vistula and Odra mouth inlfowing into southern Baltic. We have derived a new
formulea to determine the $a_{CDOM}(400)/S$ relationship that covered whole range of the
$a_{CDOM}(400)$ observed both the Baltic Sea and in Pomeranian lakes waters. The new formulea
was marked on Figure 2.a as red curve and is described by Equation 10.





$$S = 0.0213 - 0.003 \, ln[a_{CDOM}(400)]. \qquad (10)$$
The new $a_{CDOM}(400)/S$ relationship has been found much better constrained and explained
much more variance ($R^2 = 0.79$) with less uncertainty (RMSE = 0. 1%) compared to one
presented by Kowalczuk (2001).
Detailed analysis of distribution of spectral slope in the function of $a_{CDOM}(400)$
indicated that data set could be divided in respect to salinity, into two subsets: samples
characterised by salinity above 5 (mostly Baltic Sea water samples) and those with salinity
below 5, which include waters from river mouths, lakes and the Szczecin Lagoon. The
relationship between $a_{CDOM}(400)$ and $S$ derived for respective data substets were presented on
Figure 2.b and functial formulaes were given by Equation 11 (salinity > 5) and Equation 12
(salinity < 5)
$$S = 0.0206 - 0.004 \, ln[a_{CDOM}(400)] \qquad (11)$$
$$S = 0.0196 - 0.0009 \, ln[a_{CDOM}(400)]. \qquad (12)$$
Proposed approximations of $a_{CDOM}(400)/S$ relationships in two salinity ranges were
characterised by the higher explained variance ($R^2 = 0.78$ for Equation 11, and lower
$R^2 = 0.22$, for Equation 12, respectively. In both cases, the estimation uncertainty:
RSME = 0.08% for Equation 11, RSME = 0.09%, for Equation 12, respectively, were lower
compared to approximation presented by Equation 10.
*3.2. A model for approximation of CDOM light absorption spectrum from empirical*
*dependency with the chlorophyll a concentration.*
The principle bio-optical assumption on interdependencies among optically significant
water constituents in global ocean was formulated by Morel and Prieur (1977), who
introduced the concept of the Case 1 water, where variability if those constituents is to far
extent correlated with variability of phytoplankton biomass expressed as chlorophyll *a*
concentration. The Case 1 waters were mostly open oceanic waters and upwelling region at
western continental margins. The marine basins where these assumption were not fulfilled
were considered Case 2 water: mostly semi-enclosed and shelf seas and coastal ocean, where
there were sources of riverine waters. It was assumed that changes in magnitude of optically
significant water constituents in the Case 2 waters were independent. This concept was
critically reassessed by Siegel et al. (2005) who reanalyzed the global ocean colour imagery



data set and proved that, although in open ocean the bio-optical assumption is still valid, there
were significant dependences between chlorophyll *a* and other optically significant water
constituents at regional scales in oceanic continental margins. Even though the CDOM was
not thought to be correlated with chlorophyll *a* concentrations in Case 2 waters, there were
examples showing that such a relationships were possible (Ferrari and Tassan, 1992; Vodacek
et al., 1997). In the Baltic waters such analyses were carried out by Kowalczuk and
Kaczmarek (1996) and Kowalczuk (1999). These authors demonstrated that the correlation
between the concentration of chlorophyll *a* and the CDOM absorption coefficient was
observed. The positive correlation between light absorption by CDOM chlorophyll *a*
concentration has been confirmed with new data available, both in marine and fresh waters.
The clear trend of increase of CDOM absorption level with increasing phytoplankton biomass
has been presented on Figure 3. The dependence between $a_{CDOM}(400)$ coefficient and the
concentration *Chla* obtained by Kowalczuk (2001) has been overlaid on the new, currently
reported empirical data set, Figure 3. It is evident that $a_{CDOM}(400)$/*Chla* relationship reported
by Kowalczuk's is applicable to only some of the Baltic Sea data, in chlorophyll *a*
concentration range $0.8 < Chla < 10$ mg m$^{-3}$. The old, power function relationship did not
reproduced correctly the $a_{CDOM}(400)$ values for high chlorophyll *a* concentration, and CDOM
absorption data measured in estuaries and lakes were lying above the model curve. We have
proposed new statistically significant relationship between the $a_{CDOM}(400)$ and *Chla* which
was described by a second-degree polynomial (R$^2$ = 0.83, RMSE = 28%, n = 541, p<0.0001).

The same function has been applied to reconstruct the complete CDOM absorption

spectrum in the spectral range from 245 to 700 nm with 5 nm resolution, Equation 13:

$$a_{CDOM}(\lambda) = 10^{(A(\lambda)(\log Chla)^2 + B(\lambda)\log Chla + D(\lambda))}, \qquad (13)$$

where $A(\lambda)$ [m$^5$ mg$^{-2}$], $B(\lambda)$ [m$^2$ mg$^{-1}$], $D(\lambda)$ [m$^{-1}$] are the regression coefficients.

The spectral distribution of the regression coefficients and determination coefficient

have been presented on Figure 4 and their values were included in Table A in Appendix A.
Both regression coefficients $A(\lambda)$ and $B(\lambda)$ showed relatively small spectral variation in the
UV and part of the visible spectral range. The biggest changes in regression coefficients
spectra have been noted above 580 nm, where significant increase of the $A(\lambda)$ has been
relatively compensated with decrease of the $B(\lambda)$. Spectral distribution of regression
coefficient *A*, indicated a potential influence of the phytoplankton pigments absorption on the
CDOM absorption spectrum as its maximum situated around 675 nm, overlaps with long



wave maximum of chlorophyll *a* absorption spectrum. This effect is visible only at longer
wavelength because the principle chlorophyll *a* maximum at 440 nm, is masked by CDOM
absorption especially at very turbid estuarine and fresh water, where highest values of CDOM
absorption were recorded. Free term $D(\lambda)$ spectrum decreases monotonically with increased
wavelength resembles the of the log transformed CDOM absorption coefficient spectrum
corresponding to the average CDOM absorption spectrum at a chlorophyll *a* concentration of
1 mg m$^{-3}$ as shown on Figur 4.c. The spectral distribution of the determination coefficient
values $R^2$, presented on Figure   4.d, demonstrated that, the model based on the dependency
between CDOM absorption coefficient and chlorophyll *a* concentration, explained more than
80% variability in $a_{\text{CDOM}}(\lambda)$ in UV and VIS, and this variability was controlled by
phytoplankton biomass production. The model performance deteriorated at wavelength longer
than 550 nm.

The model uncertainty has been passed and uncertainty analysis result for selected

wavelengths have been summarized in Table 3 and presented on Figure 5. Comparison
between estimated vs. measured $a_{\text{CDOM}}(\lambda)$ values at selected wavelengths (260, 350, 440,
500, 550, 600 nm) from range 240 – 700 nm were shown on first six upper panels Figure 5 (a-
f). Histograms of ratio between estimated and measured values at the same wavelength were
presented on lower six Figure 5 panels (g-l). The deterioration of model performance with
increasing wavelength has been evident. The overall uncertainty expressed by arithmetic
statistics and logarithmic statistics is satisfactory up to 500 nm, and then both systematic and
statistical estimation errors increased rapidly at longer wavelength. The arithmetic systematic
error has increased from 1.47% at 260 nm to 19.54% at 600 nm, arithmetic statistical error has
increased from 17.03% at 260 nm, to 79.13% at 600 respectively. Logarithmic uncertainty
metrics indicated that, standard error factor estimated for the entire spectral range from 240 to
700 nm of light absorption coefficients varies from 1.19 to 2.66. This means that the statistical
logarithmic error varies from -62% to +165%. The logarithmic systematic errors in the all 240
- 700 nm range do not exceed 3%.
3.3. *An empirical model for approximation of CDOM light absorption spectrum based on*
*empirical dependency with the CDOM absorption coefficient  value at 400 nm, $a_{CDOM}(400)$.*

The exponential model for CDOM absorption requires information on two input

parameters: magnitude of CDOM absorption at reference wavelength and spectral slope
value. However, the monotonic property of CDOM absorption spectrum determines the high




level of interdependency of absorption coefficient values across considered spectral range and
allows omit the detailed information on spectral slope. The second model that we have
developed is based on the dependence of light absorption by CDOM at any given wavelength
and the CDOM absorption coefficient at wavelength 400 nm. Many authors treat this
wavelength as a reference for CDOM absorption using the exponential Equation 1 (e.g.
Kowalczuk et al., 2005; Woźniak and Dera, 2007). It was also recommended by
Sathyendranath et al. (1989) to distinguishing between dissolved organic matter absorption
from that caused by phytoplankton. In optically complex waters (Baltic Sea and the lakes),
$a_{CDOM}(400)$ makes up the large proportion of the total absorption of light in water,
(Kowalczuk, 2001; Ficek 2013).
The interdependency of spectral CDOM absorption values has been assessed by
Kowalczuk (2001) who analysed the linear cross-correlation matrix between $a_{CDOM}(\lambda)$ values
measured at different wavelengths. The linear interrelationship between $a_{CDOM}(\lambda)$
deteriorated with increasing spectral distance from reference wavelength both toward shorter
and longer wavelengths. To better reflect the non-linear property of CDOM absorption
spectrum we have used the second order polynomial model based on log transformed
$a_{CDOM}(\lambda)$ values as input variable. Calculation were performed in the spectral range 240 –
700 nm, with 5 nm resolution. The statistical analyses yielded the formula:
$$a_{CDOM}(\lambda) = 10^{(M(\lambda)(\log(a_{CDOM}(400))^2 + N(\lambda)\log(a_{CDOM}(400)) + O(\lambda))} , \qquad (14)$$
where $M(\lambda)$ [m], $N(\lambda)$ [dimensionless] and $O(\lambda)$ [m$^{-1}$] are the parameterization coefficients
shown graphically in Figure 6. Their values for the 240 – 700 nm range are listed in Table B
(in Appendix A).
The spectral shape of the regression coefficients $M(\lambda)$, $N(\lambda)$ and free term $O(\lambda)$ that
were derived for empirical model that used the $a_{CDOM}(400)$ value as independent variable,
were quite similar to spectral shape of regression coefficient and free term of the model based
on chlorophyll $a$ concentration. The regression $M(\lambda)$, and $N(\lambda)$ were also characterised by
maxima located in the red part of the light spectrum. Similarly to the first presented model,
the spectral shape of the free term $O(\lambda)$ resembled the log-transformed CDOM absorption
spectrum. The spectral distribution of the determination coefficient $R^2$ indicated that
approximation of $a_{CDOM}(\lambda)$ values based on the magnitude of the CDOM absorption at



reference wavelength was much more accurate than approximation based on chlorophyll *a*
concentration. The $R^2$ values were over 0.9 in ultraviolet part of the spectrum approaching 1,
near the reference value, and felt down below 0.8 at 560 nm.
The second model uncertainty has been passed, and uncertainty analysis result for the
same wavelengths as previously used, have been summarize at Table 4 and presented on
Figure 7 as presented in Table 3. Comparison between estimated vs. measured $a_{CDOM}(\lambda)$
values at six selected wavelengths were shown on first six upper panels Figure 7 (a-f).
Histograms of ratio between estimated and measured values at the same wavelength were
presented on lower six Figure 7 panels (g-l). The deterioration of model performance with
increasing wavelength has been much smaller than in case of CDOM absorption spectrum
approximation based on the chlorophyll *a* concentration. The overall uncertainty expressed by
arithmetic statistics and logarithmic statistics was much better up to 550 nm. Similarly, to the
first model both systematic and statistical estimation errors increased at longer wavelength.
The arithmetic systematic error has increased from 0.38% at 260 nm to 16.64% at 600 nm,
arithmetic statistical error has increased from 9.11% at 260 nm, to 67.45% at 600 nm
respectively. Logarithmic uncertainty metrics indicated that, standard error factor estimated
for the entire spectral range from 240 to 700 nm of light absorption coefficients varies from
1.09 to 1.76. This means that the statistical logarithmic error varies from -43% to +75%. The
systematic errors in the 240 - 700 nm interval did not exceed 2%
*3.4 Two-parametrical model for estimating of CDOM absorption in the Baltic Sea and*
*Pomeranian Lakes*
Earlier we showed two alternative one-parameter models of CDOM absorption which
estimating values of $a_{CDOM}(\lambda)$ at different wavelengths with relatively low errors. However,
there is a two-parameter model, developed by Kowalczuk et al. (2006) for the Baltic Sea
waters, which we decided to analyzed in this study data for comparison. This statistical model
for estimation of CDOM absorption coefficient at 375 nm, $a_{CDOM}(375)$ at surface waters was
based on the seasons and the chlorophyll *a* concentration that acted as a proxy for
autochthonous production of CDOM. We have used the non-linear relationship between the
CDOM absorption coefficient $a_{CDOM}(375)$ and spectral slope to derive S, and used it later for
CDOM absorption spectrum reconstruction using classical exponential model (Equation 1).



The dependence between slope $S$ and $a_{\text{CDOM}}(375)$ coefficient obtained by Kowalczuk et al.
(2006) has been overlaid on the currently reported empirical data set, Figure 8. The
$S/a_{CDOM}(375)$ relationship reported by Kowalczuk et al., (2006) is applicable to most of the
Baltic Sea and lakes data in within the $a_{\text{CDOM}}(375)$ range $1.5 - 14.16$ m$^{-1}$ (mainly estuaries
and lakes waters). That hyperbolic relationship did not reproduced correctly the $S$ values for
$a_{CDOM}(375)$ values below 1.5 m$^{-1}$, and slopes measured in open and mostly coastal Baltic
waters were lying below the model curve. We have proposed similar hyperbolic statistically
significant relationship between the $S$ and $a_{\text{CDOM}}(375)$ which could better fit to current data
set. The determination coefficient of update hyperbolic function was very high: $R^2 = 0.86$,
RMSE = 0.08%, n = 541, p<0.0001. The new empirical relationship between spectral slope $S$,
and $a_{\text{CDOM}}(375)$ is given by formula (15):
$$S = 0.01722 + \frac{0.0057}{0.0407 + a_{CDOM}\ (375)}. \qquad (15)$$

The new formulae was applied Equation 1 to calculate the CDOM absorption
spectrum in the spectral range between 240 - 700 nm. The uncertainty of the exponential
model that used the spectral slope variable estimated from the new approximation given by
Eqaution 15 have been assed and uncertainty analysis result for selected wavelengths have
been summarize at Table 5. For comparison we have also done uncertainty analysis of the
exponential model with spectral slope variable estimated from the $S$ and $a_{\text{CDOM}}(375)$
relationships presented by Kowalczuk et al. (2006). The uncertainty analysis has revealed that
two parameter estimation of the CDOM absorption spectrum was less accurate that two first
one parameters models. The spectral values of CDOM absorption estimated form the
exponential relationship and spectral slope parameterization with use of Kowalczuk et al.
(2006) and current empirical formulas were systematically overestimated in UV and
underestimated in visible spectral range. The systematic and statistical errors were increasing
toward the red part of the spectrum. The highest uncertainty, that exceeded 30% in systematic
error and 20% in statistical error were noted at wavelengths longer than 500 nm. Use of
current empirical spectral slope parameterization enables estimation of spectral $a_{\text{CDOM}}(\lambda)$
values with relatively lower errors, compared to results given by the same approach with use
of Kowalczuk et al. (2006) slope parameterization.




## 4. Discussion

Presented dataset part of the above 25 timer series of bio-optical data collected in IOPAN in the Baltic Sea. This subset was created to match the observations conducted in the 2006 - 2009 in Pomeranian lakes by Ficek et al., (2012) and Ficek (2013) and enabled extended analysis on data characterized by large dynamic variability range that in some cases exceeded three orders of magnitude. Seawaters and lake waters were analyzed as a single database, despite some differences in the compositions of optically active components of these waters, treating the lakes as a natural extension of marine waters with properties resembling the properties of estuaries. Coefficients of $a_{CDOM}(\lambda)$ in analyzed waters varies in 3 orders of magnitude (for example $a_{CDOM}(375)$ varies from 0.41 to 14.16 m$^{-1}$, $a_{CDOM}(400)$ varies from 0.15 to 8.85 m$^{-1}$). Spectral slope $S_{300-600}$ in Baltic and lakes varies in range 0.007 – 0.03 nm$^{-1}$, while the *Chla* concentration varied in range up to 3 orders of magnitude from 0.72 to 119 mg m$^{-3}$. Ranges of variability of parameters analyzed in this paper corresponded with the data presented in earlier work on the optical properties in the Baltic Sea (Babin et al. 2003, Kowalczuk 1999, Kowalczuk et al. 2005, 2006, 2010) or Pomeranian lakes (Ficek et al. 2012; Ficek 2013). Ficek (2013) reported that in Pomeranian lakes *Chla* concentrations can may be even 336 mg m$^{-3}$.

In this paper we have presented two single-parameter models and one two-parameter model, which we use for calculation spectral values CDOM absorption coefficients $a_{CDOM}(\lambda)$ in a broad spectral range in Baltic Sea waters and the Pomeranian lakes. First two models based on one single independent variable were characterized by similar uncertainty level, which was in order of 1.5 - 7% in UV and visible spectral range, when chlorophyll *a* was used as the independent variable or in order of 0.4 -2.2 %, in the same spectral range when $a_{CDOM}(400)$ was used as independent variable. For example, the statistical errors listed in Table 3 for the parameterization dependent on the chlorophyll *a* concentration (13) and in Table 4 for model (14) shows that the statistical arithmetic error is higher in the former case – e.g. for 440 nm it is 4.01% – whereas in the latter case it is 0.42%. The second one parameter model was characterized by lower uncertainty and higher spectral values of the determination coefficient. Likewise, the standard error factor in the first model is higher than in the one based on the dependence of the absorption $a_{CDOM}(\lambda)$.

The accuracy of both models have deteriorated at wavelength longer than 550 nm. One possible explanation is the precision of the CDOM measurements. The use of 5 cm cuvettes



allowed the reliable CDOM absorption detection at $a_{CDOM}(\lambda)$ larger than 0.046 m$^{-1}$. The
spectrophotometer detection limit has been reached usually at wavelengths longer than 550
nm. Therefore the modeled values were usually compared to measured values that were
heavily impacted by errors resulted from measurements accuracy. One of the possible way of
increasing the spectrophotometric accuracy of CDOM absorption measurements would rely
on increasing cuvetts pathlength (maximum cuvettes pathlength used in most desktop
spectrophotometers would not exceed 10 cm), or use the optical waveguide
spectrophotometer systems that offer the optical pathlength in range of 0.2 meter to 2 meters
by (D'Sa et al., 1999; Miller et al. 2002). However usage of long measurements pathlength in
optically complex water such as Baltic Sea and fresh water lakes would severely impact the
radiometric sensitivity of any spectrophotometer, causing the fast decrease of light intensity
reaching the detector especially in the UV spectral range.

There were number of regional studies presenting the dependence between

chlorophyll $a$ concentration, $Chla$ and CDOM absorption $a_{CDOM}(\lambda)$, similar to our
parameterization described by Equation 13 (Ferrari and Tassan, 1992, Tassan 1994, Vodacek
et al. 1997, Morel et al. 2007, Morel and Gentili 2009, Bricaud et al. 2010, Organelli et al.
2014). We have compared the $a_{CDOM}(\lambda)/Chla$ relationship derived by us with selected
relationships between CDOM absorption coefficients $a_{CDOM}(\lambda)$ and $Chla$ concentrations for
selected wavelengths developed by different authors for different water types. Selected model
outputs  were overlaid on the observed distribution of $a_{CDOM}(\lambda)$  in the function of $Chla$,
presented on Figure 9. These relationships in all cases were approximated by power functions,
and assumed different rates of increase of the $a_{CDOM}(\lambda)$ value with increasing $Chla$ (Tassan,
1994; Morel et al., 2007; Morel and Gentili 2009; Bricaud et al. 2010). Relationships
developed by other authors, were not suitable for estimating CDOM absorption in the Baltic
Sea waters and lakes. Empirical relationships developed by Tassan, (1994), Morel et al.,
(2007), Morel and Gentili (2009) and Bricaud et al. (2010) all underestimated the CDOM
absorption in Baltic Sea. Such a large mismatch between estimated and observed CDOM
absorption values certainly resulted from the fact that these relationships were developed for
clean oceanic waters where the contribution of dissolved organic material to the total light
absorption light was lower than in the Baltic Sea and the concentration of $Chla$ did not exceed
40 mg m$^{-3}$. For example, Bricaud et al. (2010) have based their empirical model on
measurements from mesotrophic waters around the Marquesas Islands to hyperoligotrophic
waters in the subtropical gyre and eutrophic waters in the upwelling area west off Chilean



coast (South Pacific), where observed *Chla* concentrations spanned more than two orders of magnitude (0.017 to 1.5 mg m$^{-3}$) in the surface layer; observed values of the spectral slope, *S*, contained within the range of 0.007 - 0.032 nm$^{-1}$; and observed $a_{CDOM}(440)$ values were from 0.0003 to 0.038 m$^{-1}$. Morel et al. (2007) carried out measurements in hyperoligotrophic waters in the South Pacific gyre (near Easter Island), where observed *Chla* concentrations were within range of 0.022 to 0.032 mg m$^{-3}$ in the surface layer. Tassan (1994) reported two relationships between $a_{CDOM}(\lambda)$ and *Chla* (one for Gulf of Naples waters and second for Adriatic Sea), and then used these relationships for estimation of CDOM absorption coefficient values at different ranges of *Chla* concentrations (0.25 do 40 mg m$^{-3}$). Morel and Gentili (2009) tested developed satellite ocean color algorithm enabling determination of the CDOM absorption and the *Chla* concentration from satellite imagery in Mediterranean waters, where *Chla* varied within the range from 0.01 to 0.5 mg m$^{-3}$. The eutrophic Baltic Sea waters and superthrophic lakes water were characterized by high significantly higher *Chla* concentrations. The total absorption in our study area were dominated by absorption organic dissolved substances (Woźniak et al., 2011; Ficek et al., 2012), which has of both the autochthonous and allochthonous origin, therefore observed spectral CDOM absorption coefficient valued per unit of chlorophyll a concentrations were almost twice as much in the Baltic Sea and Pomeranian lake than those observed in oceanic waters in the Pacific and marine water in the Mediterranean and Adriatic. These findings underlined need for development of regional algorithms and bio-optical models, because those developed in other regions did not accounted the constant and very high background in CDOM absorption persistently prevalent in the Baltic Sea and fresh water in temperate climatic zone.

The uncertainty analysis proved that, both mathematical one-parametrical CDOM absorption estimations presented in this paper performed better, than classical exponential model with variable slope non-linear parameterization by Kowalczuk et al. (2006) and its modification presented in Equation 15. Comparison of tables 3, 4 and 5 showed, that in any case, the estimation accuracy decreases with the wavelength, however the two parameters exponential model significantly underestimated $a_{CDOM}(\lambda)$ at longer wavelengths. Standard error factor *x* (indicating how many times approximated values were different from measured) is lower in the Kowalczuk et al. (2006) model and our modification of this model than approximations (13) and (14). But systematic errors, both arithmetic and logarithmic, are much higher. For example in models by Kowalczuk et al. (2006) for the 440 nm wavelength arithmetic systematic error takes average value -16% and logarithmic systematic error takes



average value -17%, while using the formula (13), we have 4% and 0.01%, and for the
formula (14) 0.4% and 0.003%, respectively. Morel and Gentili (2009) and Morel et al.
(2010) derived a two-component model for description of the CDOM absorption properties,
and they modelled the spectral slope values using its empirical relationship with the
chlorophyll $a$ concentration. These models were based on data sets collected in clear oceanic
waters, so their applicability to Baltic Sea conditions would probably be questionable as it
was in case of the $a_{CDOM}(\lambda)/Chla$ relationships.
Finally, we have compared the performance in the retrieval of CDOM absorption spectrum in
the Baltic Sea conditions of two standard exponential models broadly used in optical
oceanography: $i)$ model by Bricaud et al. (1981) with spectral slope $S_{375-500}$ and CDOM
absorption reference wavelength $\lambda_0 = 375$ nm, $ii)$ model by Babin et al. (2003) with spectral
slope, $S_{350-500}$ and CDOM absorption reference wavelength $\lambda_0 = 443$ nm) and model
Kowalczuk et al. (2006). The modelled spectra were presented on Figure 10, together with
measured CDOM absorption spectra and those calculated from one-parameter models
expressed by Equations 13 and 14. Empirical model developed for Baltic Sea and inland
waters - Equations 13 and 14, based on locally observed variability in biogeochemical and
optical variables adequately reflected the real, spectrophotometrically measured light
absorption coefficients. The model based on the dependence of the chlorophyll $a$
concentration, Equations 13, best fits the coefficients for wavelengths from 240 to 600 nm,
and could applied in variety of water bodies with contrasting trophic status. From this point of
view, therefore, it is far superior to the models derived by Bricaud et al. (1981) or Babin et al.
(2003), which were developed either for oligotrophic or mesotrophic oceanic waters, or for
European coastal water but with incorporating bio-optical properties of fresh waters. On the
other hand model Kowalczuk et al. (2006) underestimates values of $a_{CDOM}(\lambda)$.
In order to compare the above-mentioned models, we adapted them to the empirical
data set presented in this study within the spectral range from 240 to 700 nm, and then we
have applied the same statistical metrics to assess their uncertainty. Calculated errors were
listed in Table 6 for selected wavelengths. The systematic errors in arithmetic statistics were
higher in selected error compared to one parameters models presented by us. The systematic
errors calculated for CDOM absorption model by Babin et al., (2003) were significantly
higher in all selected wavelengths compared to those presented in Tables 3 and 4. The CDOM
absorption could be estimated by empirical model based on the a the $a_{CDOM}(\lambda)/Chla$



dependency with the systematic error of 3.13 % at λ = 350 nm, whereas Babin et al., (2003)
model estimated the CDOM absorption at the same wavelength with systematic error of -
33.70%. Calculated statistical errors of the estimates with use of the Bricaud et al. (1981) and
Babin et al. (2003) models were very large compared to the results obtained with models
expressed by Equations 13 and 14. Whereas the standard error factors are quite good for
Bricaud's model (from 1 to 2.43), they are much higher for Babin's model (from 1.045 to
3.58). However, in both cases, the systematic errors are large: -59% to 144–and 79% to
+400%, respectively.
**5. Conclusion**
We have demonstrated that CDOM absorption was non-linearly correlated with
chlorophyll $a$ concentration in broad variability range spanning over three orders of
magnitude in marine waters of the Baltic Sea, its estuaries, coastal lagoons and in the fresh
water lakes characterised by different throphic status. The second order polynomial
approximation the relationship between chlorophyll a concentration and $a_{CDOM}(400)$ could be
used to in both marine and fresh water, and was much more accurate than one derived for
Baltic Sea waters by Kowalczuk (2001). This relationship has also proved that optical and
bio-optical properties of marine and fresh waters could be regards as an continuum in regard
of CDOM absorption and chlorophyll $a$ concentration. We have had derived models for
estimation of CDOM light absorption by spectrum in the spectral range 240-700 nm from
chlorophyll $a$ concentrations $Chla$ or from coefficients of light absorption by CDOM for
wavelength 400 nm ($a_{CDOM}(400)$). For comparison we have also, tested the classical
exponential model for approximation CDOM absorption spectrum, where the spectral slope
coefficient was determined from nonlinear relationship between spectral slope coefficient and
values of $a_{CDOM}(375)$. The uncertainty analysis results proved that, the one-parametric,
second order polynomial function of the chlorophyll $a$ concentration, $Chla$, enabled
estimation of spectral values of CDOM absorption coefficient, $a_{CDOM}(\lambda)$ with slightly lower
accuracy than, its estimation based on second order polynomial function of the CDOM
absorption coefficient at wavelength 400 nm $a_{CDOM}(400)$. Presented models, optimized for
Baltic Sea and fresh water specific optical and bio-optical conditions, were characterized with
significantly lower errors of estimations compared to widely used CDOM absorption model
proposed by other authors. The CDOM absorption models presented in this study, could be
used for improvements of remote sensing algorithms designed for retrievals of various optical





and bio-optical parameters needed for characterization and monitoring of the state and functioning of the Baltic Sea and Pomeranian lakes ecosystems. Validation of these models showed that they can be reliably applied in monitoring surveys, when a rapid, approximation the light absorption spectrum is needed.

**Acknowledgements**

This paper was prepared as a part of project N N306 041136 financed by the Polish Ministry of Science and Higher Education in 2009-2014 and also within the framework of the SatBałtyk project funded by the European Union through the European Regional Development Fund, (No. POIG.01.01.02-22-011/09, 'The Satellite Monitoring of the Baltic Sea Environment'). The Institute of Oceanology, Polish Academy of Sciences provided the funds for part of this study within the framework of the Statutory Research Project.

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





**Table 1**. Dates, number of samples collected and parameters measured during cruises and field experiments made for this study.

| Dates of cruises | Number of samples | Parameters measured | Region |
|---|---|---|---|
| 24-31 Aug. 2006 | 20 | $a_{CDOM}(\lambda)$, *Chla*, CTD | southern Baltic Proper, Gulf of Gdańsk |
| 24-29 Sept. 2006 | 12 | $a_{CDOM}(\lambda)$, *Chla*, CTD | southern Baltic Proper, Gulf of Gdańsk |
| 18-28 Oct. 2006 | 30 | $a_{CDOM}(\lambda)$, *Chla*, CTD | southern Baltic Proper, Gulf of Gdańsk, Pomeranian Bay |
| 21-31 March 2007 | 36 | $a_{CDOM}(\lambda)$, *Chla*, CTD | southern Baltic Proper, Gulf of Gdańsk, Pomeranian Bay, Szczecin Lagoon |
| 21-31 May 2007 | 38 | $a_{CDOM}(\lambda)$, *Chla*, CTD | southern Baltic Proper, Gulf of Gdańsk |
| 20-28 Oct. 2007 | 26 | $a_{CDOM}(\lambda)$, *Chla*, CTD | southern Baltic Proper, Gulf of Gdańsk |
| 01-11 March 2008 | 29 | $a_{CDOM}(\lambda)$, *Chla*, CTD | southern Baltic Proper, Gulf of Gdańsk, Pomeranian Bay |
| 11-18 April 2008 | 22 | $a_{CDOM}(\lambda)$, *Chla*, CTD | southern Baltic Proper, Gulf of Gdańsk |
| 06-14 May 2008 | 23 | $a_{CDOM}(\lambda)$, *Chla*, CTD | southern Baltic Proper, Gulf of Gdańsk |
| 01-09 Sept. 2008 | 26 | $a_{CDOM}(\lambda)$, *Chla*, CTD | southern Baltic Proper, Gulf of Gdańsk, Pomeranian Bay, Szczecin Lagoon |
| 25-29 Nov. 2008 | 18 | $a_{CDOM}(\lambda)$, *Chla*, CTD | Gulf of Gdańsk |
| 04-12 March 2009 | 14 | $a_{CDOM}(\lambda)$, *Chla*, CTD | Gulf of Gdańsk, Gotland Basin |
| 15-21 April 2009 | 29 | $a_{CDOM}(\lambda)$, *Chla*, CTD | southern Baltic Proper, Gulf of Gdańsk |
| 20-28 May 2009 | 34 | $a_{CDOM}(\lambda)$, *Chla*, CTD | southern Baltic Proper, Gulf of Gdańsk, Pomeranian Bay, Szczecin Lagoon |
| 07-16 Sept. 2009 | 35 | $a_{CDOM}(\lambda)$, *Chla*, CTD | southern Baltic Proper, Gulf of Gdańsk |
| 06-10 Oct. 2009 | 21 | $a_{CDOM}(\lambda)$, *Chla*, CTD | southern Baltic Proper, Gulf of Gdańsk |
| Dec. 2006 – Sept. | 66 | $a_{CDOM}(\lambda)$, *Chla* | Sopot Pier |
| April – Dec. 2007 | 10 | $a_{CDOM}(\lambda)$, *Chla* | Lake Łebsko |



| | | | |
|---|---|---|---|
| April – Sept. 2008 | 8 | $a_{CDOM}(\lambda)$, *Chla* | Lake Łebsko |
| June – Oct. 2009 | 9 | $a_{CDOM}(\lambda)$, *Chla* | Lake Łebsko |
| March – Dec. 2007 | 10 | $a_{CDOM}(\lambda)$, *Chla* | Lake Chotkowskie |
| Feb. – Sept. 2008 | 8 | $a_{CDOM}(\lambda)$, *Chla* | Lake Chotkowskie |
| April – Nov. 2009 | 8 | $a_{CDOM}(\lambda)$, *Chla* | Lake Chotkowskie |
| March – Dec. 2007 | 9 | $a_{CDOM}(\lambda)$, *Chla* | Lake Obłęskie |
| Feb. – Sept. 2008 | 8 | $a_{CDOM}(\lambda)$, *Chla* | Lake Obłęskie |
| May – Nov. 2009 | 7 | $a_{CDOM}(\lambda)$, *Chla* | Lake Obłęskie |
| All data | 556 | | |




**Table 2**. Range of variability of the spectral slope $S$, the coefficient of light absorption by
CDOM for wavelengths $\lambda = 375$ nm and 400 nm, $a_{CDOM}(375)$ and $a_{CDOM}(400)$, and
concentrations of chlorophyll $a$, *Chla*, calculated for the empirical data analysed here.

| Study area | range of variability | mean value | SD |
|---|---|---|---|
| $S$ [nm$^{-1}$] | | | |
| Baltic | 0.014 – 0.03 | 0.022 | 0.0021 |
| lakes | 0.007 – 0.02 | 0.017 | 0.0030 |
| together | 0.007 – 0.03 | 0.021 | 0.0022 |
| $a_{CDOM}(375)$ [m$^{-1}$] | | | |
| Baltic | 0.41 – 7.92 | 1.61 | 1.17 |
| lakes | 2.11 – 14.16 | 7.11 | 3.36 |
| together | 0.41 – 14.16 | 2.06 | 2.17 |
| $a_{CDOM}(400)$ [m$^{-1}$] | | | |
| Baltic | 0.15 – 4.79 | 0.997 | 0.73 |
| lakes | 1.28 – 8.85 | 4.47 | 2.07 |
| together | 0.15 – 8.85 | 1.35 | 1.41 |
| *Chla* [mg m$^{-3}$] | | | |
| Baltic | 0.72 – 76.94 | 8.77 | 11.61 |
| lakes | 1.48 – 118.97 | 39.11 | 34.15 |
| together | 0.72 – 118.97 | 13.09 | 19.78 |





**Table 3.** Relative errors of empirical model expressed by formula (13) enabling the
determination of spectral values of CDOM absorption coefficients
($a_{CDOM}(\lambda)$) at selected wavelengths.

| Wavelength [nm] | Arithmetic statistics | | Logarithmic statistics | | | |
|---|---|---|---|---|---|---|
| | systematic error | statistical error | systematic error | standard error factor | statistical error | |
| | $\langle \varepsilon \rangle$ [%] | $\sigma_\varepsilon$ [%] | $\langle \varepsilon \rangle_g$ [%] | $x$ | $\sigma_+$ [%] | $\sigma_-$ [%] |
| 260 | 1.47 | 17.03 | 0.00 | 1.19 | 19.06 | -16.01 |
| 350 | 3.13 | 25.16 | -0.01 | 1.29 | 29.01 | -22.49 |
| 440 | 4.01 | 29.37 | -0.01 | 1.33 | 32.71 | -24.65 |
| 500 | 6.54 | 39.43 | 0.01 | 1.42 | 42.45 | -29.80 |
| 550 | 11.03 | 55.07 | 0.00 | 1.57 | 57.40 | -36.47 |
| 600 | 19.54 | 79.13 | -0.09 | 1.83 | 83.43 | -45.48 |

**Table 4**. Relative errors of empirical model expressed by formula (14) enabling the
determination of spectral values of CDOM absorption coefficients ($a_{CDOM}(\lambda)$) at
selected wavelengths.

| Wavelength [nm] | Arithmetic statistics | | Logarithmic statistics | | | |
|---|---|---|---|---|---|---|
| | systematic error | statistical error | systematic error | standard error factor | statistical error | |
| | $\langle \varepsilon \rangle$ [%] | $\sigma_\varepsilon$ [%] | $\langle \varepsilon \rangle_g$ [%] | $x$ | $\sigma_+$ [%] | $\sigma_-$ [%] |
| 260 | 0.38 | 9.11 | 0.00 | 1.09 | 8.94 | -8.21 |
| 350 | 0.20 | 6.43 | -0.01 | 1.07 | 6.86 | -6.42 |
| 440 | 0.42 | 9.51 | 0.00 | 1.09 | 9.39 | -8.59 |
| 500 | 2.21 | 22.11 | 0.01 | 1.23 | 23.01 | -18.71 |
| 550 | 6.24 | 37.86 | 0.00 | 1.42 | 41.79 | -29.47 |
| 600 | 16.61 | 67.45 | -0.01 | 1.76 | 75.88 | -43.14 |






**Table 5**. Relative errors of empirical model expressed by formula dependence (15) and (1) enabling determination of spectral values of CDOM absorption coefficients ($a_{CDOM}(\lambda)$) at selected wavelengths.

| Wavelength [nm] | Arithmetic statistics | | Logarithmic statistics | | | |
| --- | --- | --- | --- | --- | --- | --- |
| | systematic error | statistical error | systematic error | standard error factor | statistical error | |
| | $\langle \varepsilon \rangle$ [%] | $\sigma_\varepsilon$ [%] | $\langle \varepsilon \rangle_g$ [%] | $x$ | $\sigma_+$ [%] | $\sigma_-$ [%] |
| 260 | 2.81 | 14.14 | 1.82 | 1.15 | 15.33 | -13.29 |
| 350 | 3.69 | 4.46 | 3.59 | 1.04 | 4.49 | -4.30 |
| 440 | -14.74 | 14.13 | -15.86 | 1.18 | 17.53 | -14.92 |
| 500 | -31.15 | 22.06 | -34.44 | 1.37 | 36.54 | -26.76 |
| 550 | -43.73 | 31.25 | -50.93 | 1.67 | 67.41 | -40.27 |
| 600 | -36.05 | 50.48 | -50.16 | 2.01 | 101.01 | -50.25 |
| Kowalczuk et al. 2006 | | | | | | |
| 260 | 9.32 | 11.48 | 8.62 | 1.13 | 13.02 | -11.52 |
| 350 | 5.14 | 4.70 | 5.04 | 1.05 | 4.68 | -4.47 |
| 440 | -18.16 | 13.96 | -19.29 | 1.18 | 17.90 | -15.18 |
| 500 | -35.34 | 21.93 | -38.71 | 1.38 | 38.23 | -27.66 |
| 550 | -47.27 | 27.17 | -53.46 | 1.65 | 64.71 | -39.29 |
| 600 | -41.25 | 46.17 | -54.77 | 2.05 | 104.97 | -51.21 |






**Table 6.** Relative errors of Bricaud et al. (1981) and Babin et al. (2003) models enabling determination of spectral values of CDOM absorption coefficients ($a_{CDOM}(\lambda)$) at selected wavelengths.

| Wavelength [nm] | Arithmetic statistics | | Logarithmic statistics | | | |
|---|---|---|---|---|---|---|
| | systematic error | statistical error | systematic error | standard error factor | statistical error | |
| | $\langle \varepsilon \rangle$ [%] | $\sigma_\varepsilon$ [%] | $\langle \varepsilon \rangle_g$ [%] | $x$ | $\sigma_+$ [%] | $\sigma_-$ [%] |
| Bricaud et al. 1981 | | | | | | |
| 260 | -35.74 | 20.98 | -38.79 | 1.36 | 35.97 | -26.46 |
| 350 | -6.95 | 3.64 | -7.02 | 1.04 | 3.98 | -3.82 |
| 440 | 11.10 | 8.51 | 10.78 | 1.08 | 7.95 | -7.37 |
| 500 | 14.24 | 19.13 | 12.82 | 1.17 | 16.72 | -14.32 |
| 550 | 11.21 | 30.85 | 7.70 | 1.28 | 27.77 | -21.74 |
| 600 | 51.80 | 90.23 | 33.10 | 1.64 | 64.00 | -39.03 |
| Babin et al. 2003 | | | | | | |
| 260 | -58.45 | 27.26 | -65.30 | 1.78 | 77.78 | -43.75 |
| 350 | -33.70 | 13.85 | -35.08 | 1.23 | 22.59 | -18.43 |
| 440 | -4.69 | 4.10 | -4.78 | 1.04 | 4.45 | -4.26 |
| 500 | 12.87 | 18.23 | 11.40 | 1.18 | 17.77 | -15.09 |
| 550 | 26.12 | 42.51 | 19.30 | 1.40 | 40.12 | -28.63 |
| 600 | 92.38 | 137.52 | 55.82 | 1.95 | 95.05 | -48.73 |






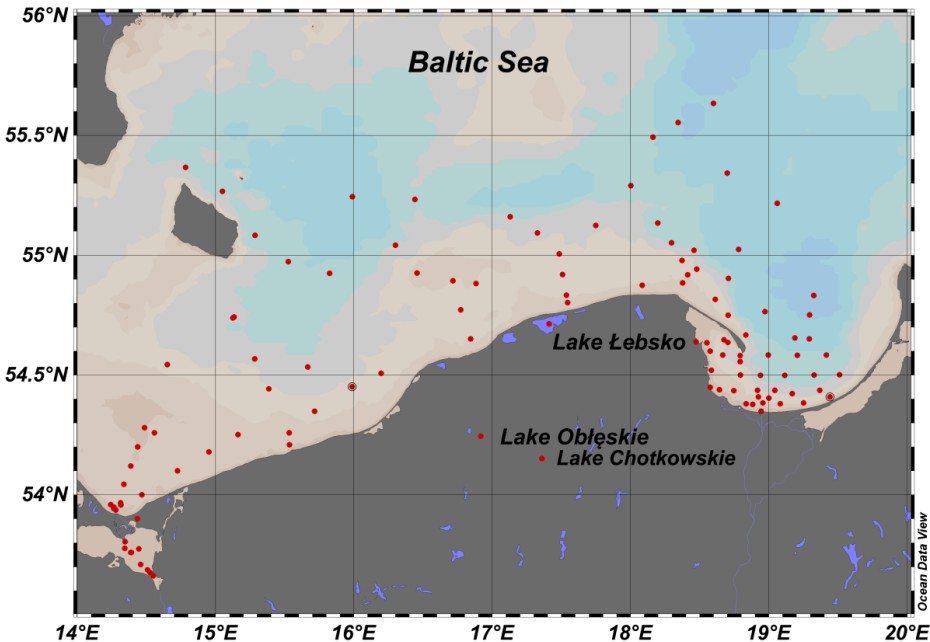


**Figure. 1.** Location of the measurement stations in 2006 – 2009





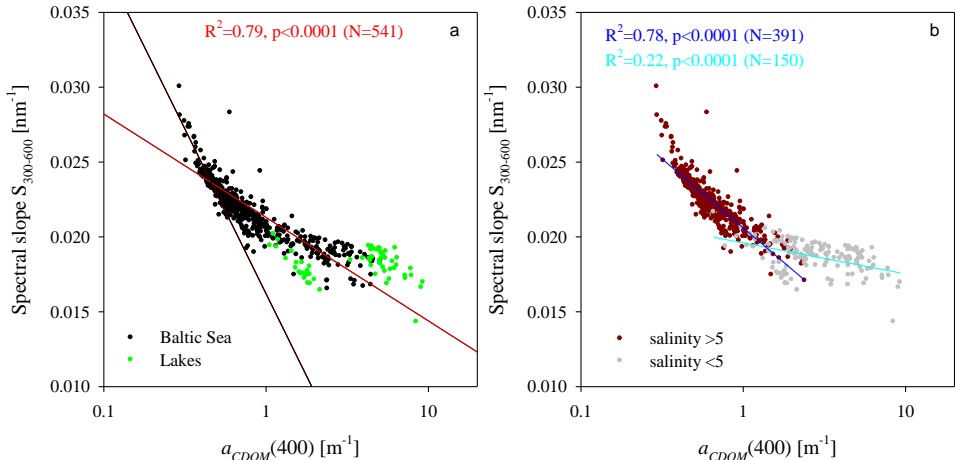


**Figure 2**. Relationship between the spectral slope $S$ and the coefficient of light absorption by CDOM for wavelength 400 nm, $a_{CDOM}(400)$ (a) in the Baltic (black dots) and lakes (green dots). The black curve is the approximation obtained by Kowalczuk (2001), the red line represent approximation expressed by Equation 10; (b) for samples with salinity above 5 (most of the sea water samples) and with salinity below 5 (samples from lakes, river mouths, the Szczecin Lagoon). The blue line represent expressed by Equation (11), and the cyan line by Equation (12).





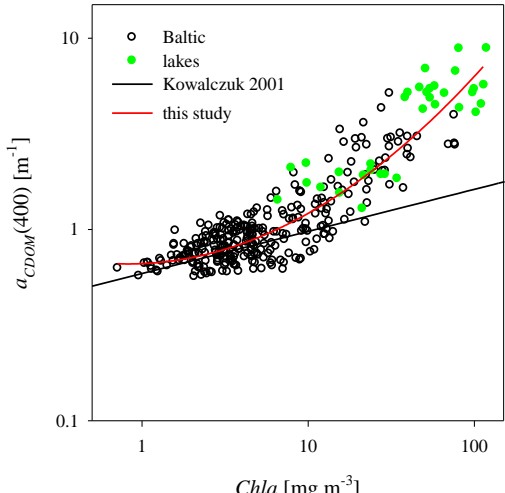

**Figure 3.** Dependence between coefficients of light absorption by CDOM $a_{CDOM}(400)$ and

chlorophyll *a* concentration. The black line shows the approximation obtained by

Kowalczuk (2001) and the red line shows approximation second-degree

polynomial in log-log scale.




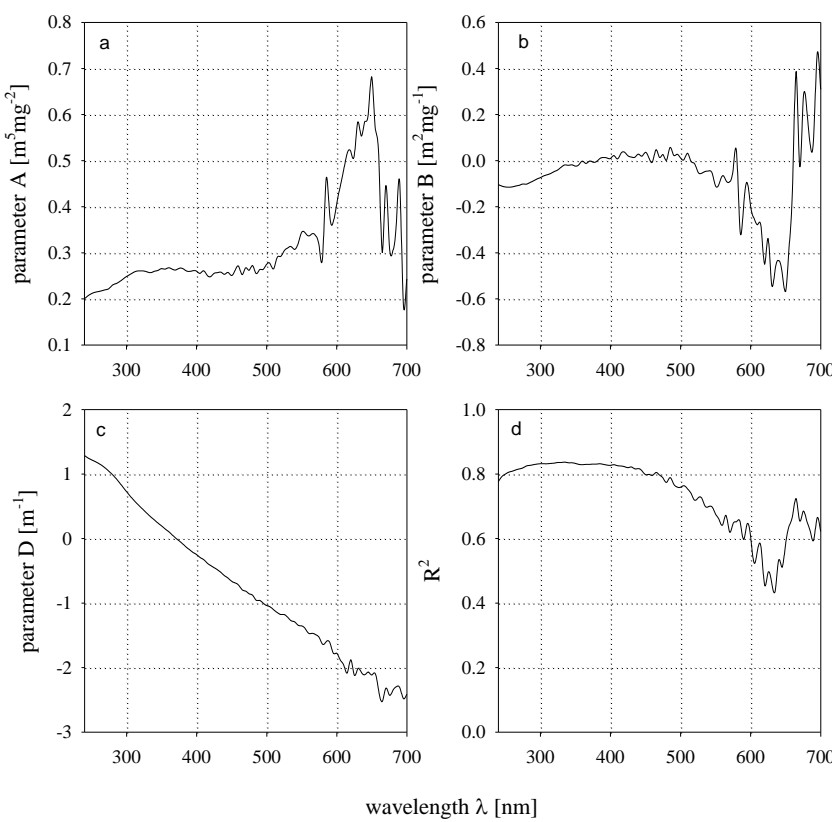

**Figure 4.** Spectral dependence of the model (expressed by Equation 13) regression coefficients (panels a and b), free term (panel c) and determination coefficient, (panel d).



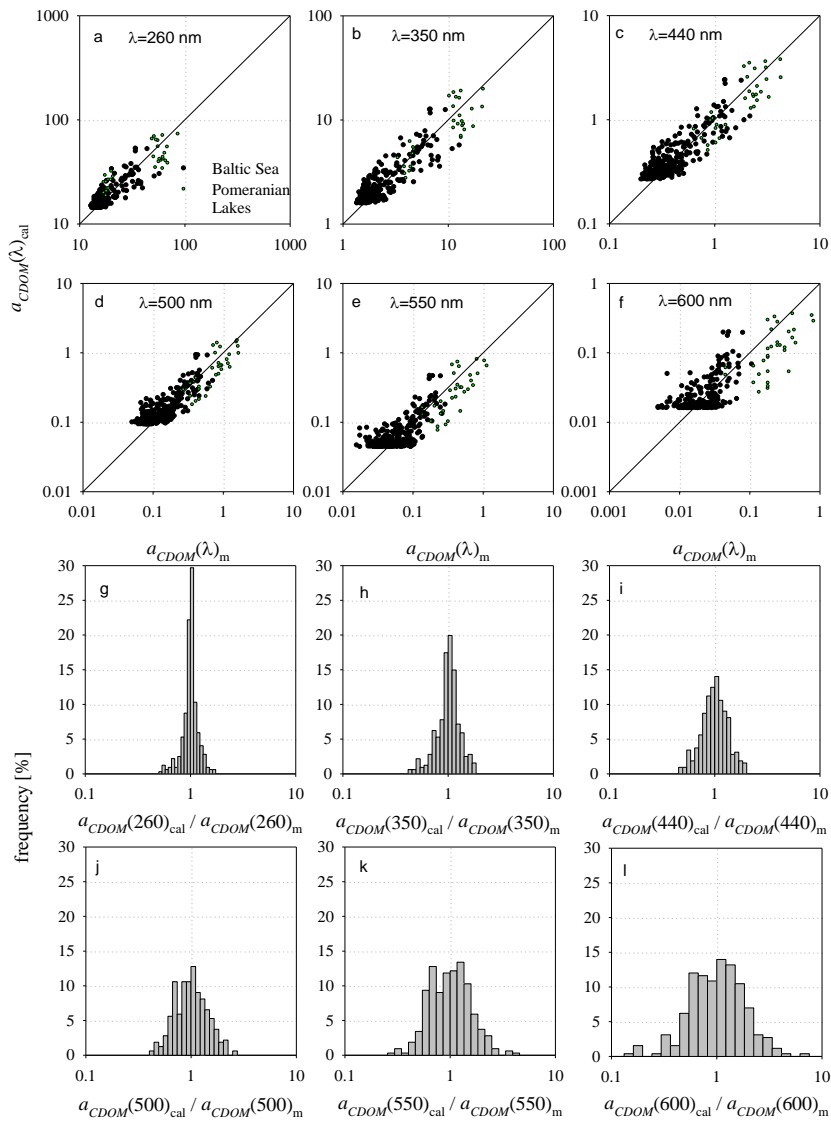

**Figure 5.** Comparison of light absorption coefficients calculated ($a_{CDOM}(\lambda)_{cal}$) using model (13) and measured ($a_{CDOM}(\lambda)_m$) in the Baltic (black dots) and Pomeranian lakes (green dots) for selected wavelengths: (a) 260 nm; (b) 350 nm; (c) 440 nm; (d) 500 nm; (e) 550 nm; (f) 600 nm. The solid line shows the function $a_{CDOM}(\lambda)_{cal} = a_{CDOM}(\lambda)_m$. And the probability density distributions of the ratio of calculated $a_{CDOM}(\lambda)_{cal}$ to measured $a_{CDOM}(\lambda)_m$ light absorption coefficients for selected wavelengths: (g) 260 nm; (h) 350 nm; (i) 440 nm; (j) 500 nm; (k) 550 nm; (l) 600 nm.




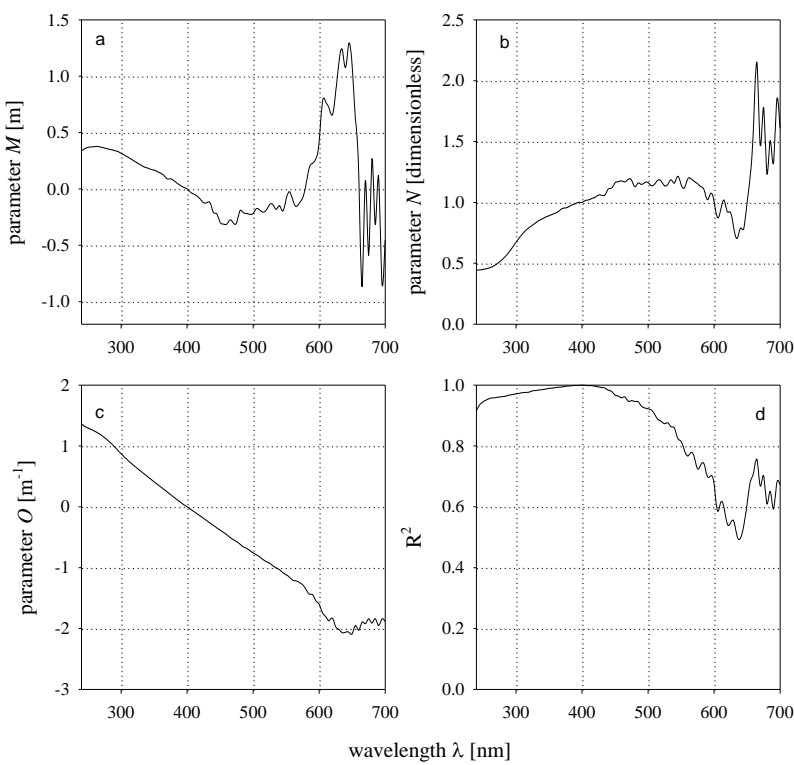

wavelength λ [nm]

**Figure 6.** Spectral dependence of the model (expressed by Equation 14) regression
coefficients ( panels a and b), free term (panel c) and determination coefficient,
(panel d).





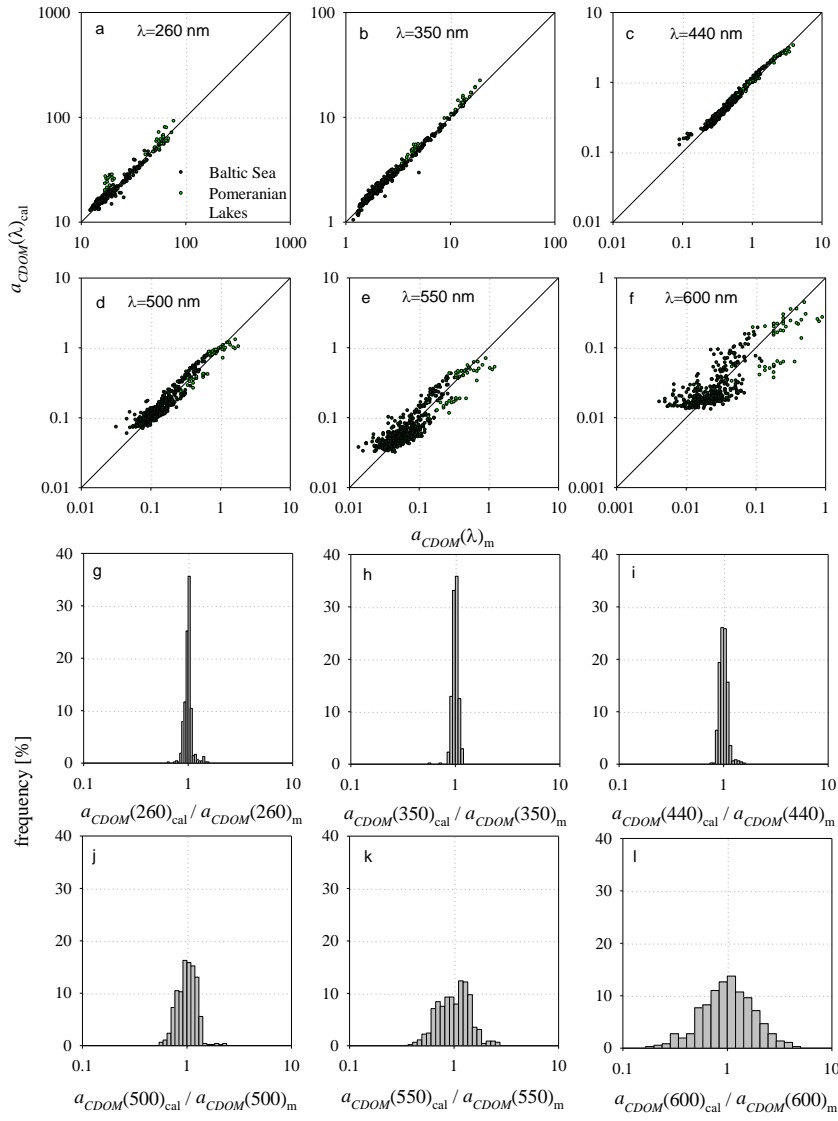

**Figure 7.** Comparison of light absorption coefficients calculated ($a_{CDOM}(\lambda)_{cal}$) using model (14) and measured ($a_{CDOM}(\lambda)_m$) in the Baltic (black dots) and Pomeranian lakes (green dots) for selected wavelengths: (a) 260 nm; (b) 350 nm; (c) 440 nm; (d) 500 nm; (e) 550 nm; (f) 600 nm. The solid line represents the function $a_{CDOM}(\lambda)_{cal} = a_{CDOM}(\lambda)_m$   And the probability density distribution of the ratio of calculated $a_{CDOM}(\lambda)_{cal}$ to measured $a_{CDOM}(\lambda)_m$ light absorption coefficients for selected wavelengths: (g) 260 nm; (h) 350 nm; (i) 440 nm; (j) 500 nm; (k) 550 nm; (l) 600 nm.



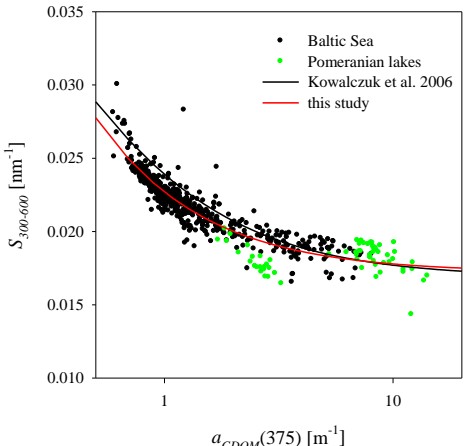


**Figure. 8.** The relationship between the spectral slope coefficient $S$, and $a_{CDOM}(375)$ in the

Baltic (black dots) and lakes (green dots). Black line indicates the model of

Kowalczuk et al. (2006) and red one indicates our new approximation (15).




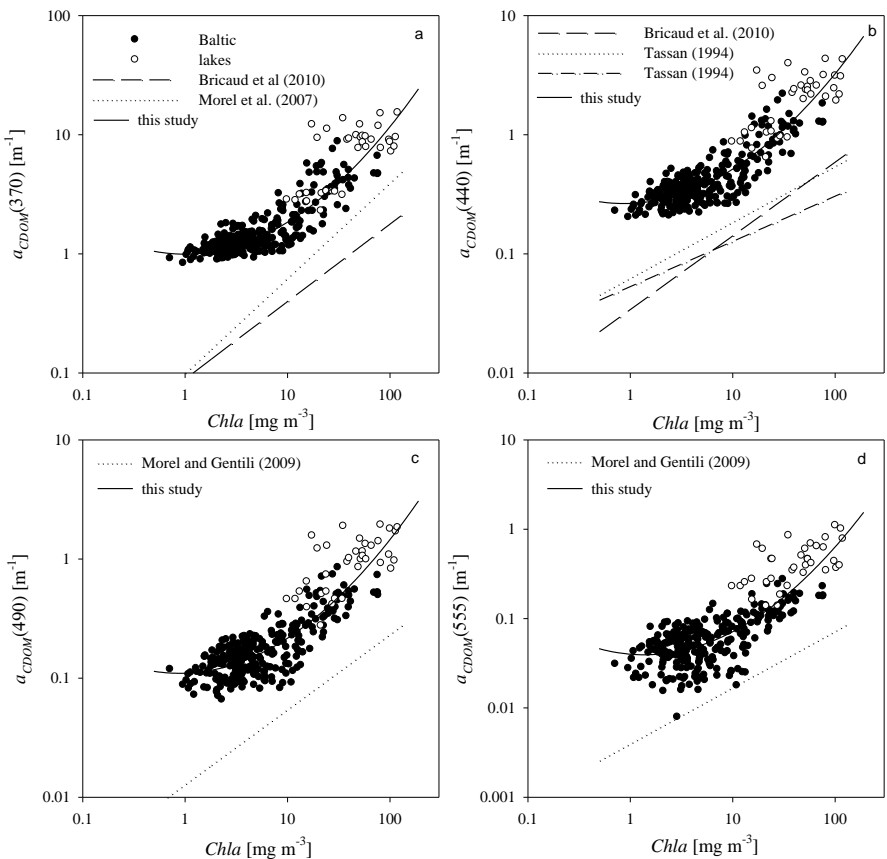


**Figure 9.** Comparison of relationships between $a_{\mathrm{CDOM}}(\lambda)$ and *Chla* developed in this work

and obtained by different authors for different waters adapted to the data analyzed

in this work.



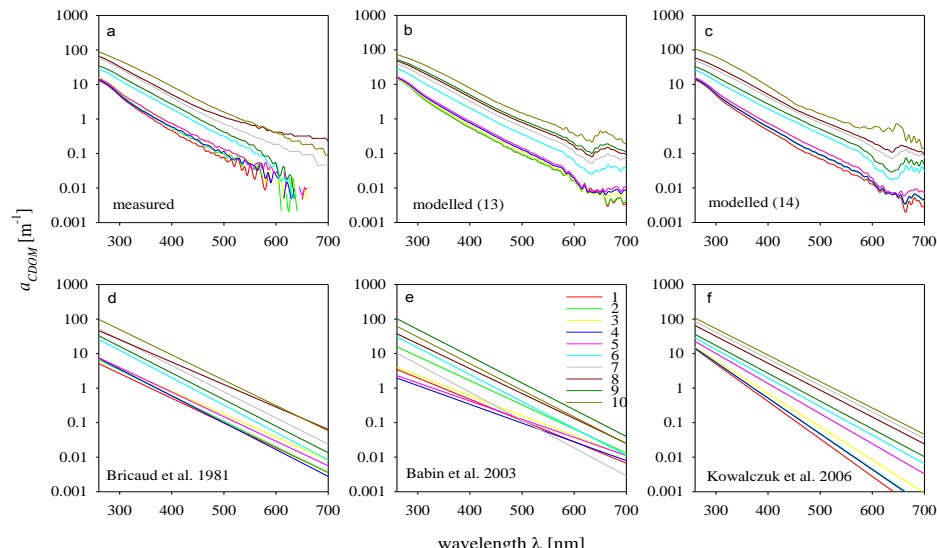


**Figure 10.** CDOM light absorption spectra: (a) empirical; (b) calculated using model (13); (c)

calculated using model (14); (d) calculated using the model of Bricaud et al. (1981);

(e) calculated using the model of Babin et al. (2003); (f) calculated using the model of

Kowalczuk et al (2006) for the following concentrations of chlorophyll *a*:

(1) $C_a = 0.96$ mg m$^{-3}$, (2) $C_a = 1.31$ mg m$^{-3}$, (3) $C_a = 3.35$ mg m$^{-3}$, (4) 4.94 mg m$^{-3}$, (5)

$C_a = 6.5$ mg m$^{-3}$, (6) $C_a = 28$ mg m$^{-3}$, (7) $C_a = 50$ mg m$^{-3}$, (8) $C_a = 66.43$ mg m$^{-3}$, (9)

$C_a = 76.95$ mg m$^{-3}$, (10) $C_a = 119$ mg m$^{-3}$





**Appendix A.**

**Table A**. Model parameters for light absorption by CDOM (13) for the wavelength range 240
- 700 nm shown for intervals of 5 nm

| wave-length [nm] | $A$ $[m^5\ mg^{-2}]$ | $B$ $[m^2\ mg^{-1}]$ | $D$ $[m^{-1}]$ | $R^2$ | wave-length [nm] | $A$ $[m^5\ mg^{-2}]$ | $B$ $[m^2\ mg^{-1}]$ | $D$ $[m^{-1}]$ | $R^2$ |
|---|---|---|---|---|---|---|---|---|---|
| *1* | *2* | *3* | *4* | *5* | *1* | *2* | *3* | *4* | *5* |
| 240 | 0.200 | -0.104 | 1.286 | 0.78 | 475 | 0.262 | 0.027 | -0.857 | 0.79 |
| 245 | 0.207 | -0.110 | 1.250 | 0.79 | 480 | 0.272 | 0.002 | -0.880 | 0.77 |
| 250 | 0.211 | -0.114 | 1.221 | 0.80 | 485 | 0.255 | 0.057 | -0.956 | 0.79 |
| 255 | 0.214 | -0.115 | 1.195 | 0.81 | 490 | 0.263 | 0.024 | -0.959 | 0.77 |
| 260 | 0.216 | -0.114 | 1.166 | 0.81 | 495 | 0.264 | 0.028 | -1.003 | 0.76 |
| 265 | 0.218 | -0.110 | 1.131 | 0.81 | 500 | 0.275 | 0.010 | -1.038 | 0.76 |
| 270 | 0.220 | -0.107 | 1.090 | 0.82 | 505 | 0.277 | 0.005 | -1.059 | 0.76 |
| 275 | 0.222 | -0.101 | 1.041 | 0.82 | 510 | 0.265 | 0.032 | -1.105 | 0.75 |
| 280 | 0.230 | -0.102 | 0.990 | 0.83 | 515 | 0.290 | -0.003 | -1.147 | 0.74 |
| 285 | 0.233 | -0.095 | 0.931 | 0.83 | 520 | 0.292 | -0.013 | -1.177 | 0.72 |
| 290 | 0.237 | -0.088 | 0.865 | 0.83 | 525 | 0.304 | -0.050 | -1.178 | 0.73 |
| 295 | 0.243 | -0.080 | 0.795 | 0.83 | 530 | 0.310 | -0.055 | -1.221 | 0.73 |
| 300 | 0.249 | -0.074 | 0.727 | 0.83 | 535 | 0.313 | -0.047 | -1.275 | 0.70 |
| 305 | 0.253 | -0.066 | 0.660 | 0.83 | 540 | 0.307 | -0.045 | -1.292 | 0.70 |
| 310 | 0.258 | -0.061 | 0.599 | 0.83 | 545 | 0.320 | -0.054 | -1.345 | 0.70 |
| 315 | 0.260 | -0.055 | 0.541 | 0.83 | 550 | 0.344 | -0.110 | -1.354 | 0.68 |
| 320 | 0.261 | -0.047 | 0.487 | 0.83 | 555 | 0.344 | -0.101 | -1.398 | 0.66 |
| 325 | 0.261 | -0.040 | 0.435 | 0.84 | 560 | 0.337 | -0.065 | -1.470 | 0.64 |
| 330 | 0.258 | -0.027 | 0.382 | 0.84 | 565 | 0.341 | -0.087 | -1.468 | 0.67 |
| 335 | 0.257 | -0.019 | 0.332 | 0.84 | 570 | 0.337 | -0.091 | -1.491 | 0.62 |
| 340 | 0.260 | -0.020 | 0.286 | 0.84 | 575 | 0.314 | -0.040 | -1.537 | 0.65 |
| 345 | 0.262 | -0.018 | 0.238 | 0.84 | 580 | 0.291 | 0.036 | -1.641 | 0.65 |
| 350 | 0.266 | -0.024 | 0.196 | 0.83 | 585 | 0.462 | -0.307 | -1.597 | 0.65 |
| 355 | 0.265 | -0.018 | 0.150 | 0.83 | 590 | 0.382 | -0.195 | -1.612 | 0.60 |
| 360 | 0.268 | -0.022 | 0.108 | 0.83 | 595 | 0.367 | -0.095 | -1.776 | 0.65 |
| 365 | 0.265 | -0.012 | 0.059 | 0.83 | 600 | 0.405 | -0.198 | -1.778 | 0.61 |
| 370 | 0.263 | -0.002 | 0.008 | 0.83 | 605 | 0.444 | -0.251 | -1.886 | 0.52 |
| 375 | 0.266 | -0.007 | -0.035 | 0.83 | 610 | 0.480 | -0.278 | -1.963 | 0.57 |
| 380 | 0.266 | -0.004 | -0.081 | 0.83 | 615 | 0.516 | -0.288 | -2.083 | 0.57 |
| 385 | 0.261 | 0.009 | -0.131 | 0.83 | 620 | 0.520 | -0.450 | -1.879 | 0.46 |
| 390 | 0.260 | 0.014 | -0.174 | 0.83 | 625 | 0.510 | -0.337 | -2.118 | 0.50 |
| 395 | 0.261 | 0.012 | -0.216 | 0.83 | 630 | 0.584 | -0.538 | -2.015 | 0.46 |
| 400 | 0.260 | 0.009 | -0.248 | 0.83 | 635 | 0.553 | -0.471 | -2.075 | 0.44 |
| 405 | 0.255 | 0.022 | -0.294 | 0.83 | 640 | 0.585 | -0.434 | -2.110 | 0.53 |
| 410 | 0.261 | 0.008 | -0.326 | 0.83 | 645 | 0.600 | -0.487 | -2.069 | 0.51 |



| 415 | 0.252 | 0.032 | -0.379 | 0.83 | 650 | 0.682 | -0.567 | -2.115 | 0.59 |
|-----|-------|-------|--------|------|-----|-------|--------|--------|------|
| 420 | 0.248 | 0.037 | -0.418 | 0.82 | 655 | 0.572 | -0.371 | -2.096 | 0.64 |
| 425 | 0.255 | 0.021 | -0.451 | 0.82 | 660 | 0.512 | -0.099 | -2.375 | 0.67 |
| 430 | 0.257 | 0.016 | -0.486 | 0.82 | 665 | 0.301 | 0.387 | -2.524 | 0.72 |
| 435 | 0.258 | 0.015 | -0.529 | 0.82 | 670 | 0.446 | -0.024 | -2.320 | 0.66 |
| 440 | 0.253 | 0.028 | -0.577 | 0.82 | 675 | 0.319 | 0.264 | -2.428 | 0.69 |
| 445 | 0.258 | 0.019 | -0.614 | 0.81 | 680 | 0.305 | 0.224 | -2.352 | 0.66 |
| 450 | 0.251 | 0.036 | -0.662 | 0.80 | 685 | 0.360 | 0.072 | -2.297 | 0.62 |
| 455 | 0.262 | 0.011 | -0.688 | 0.80 | 690 | 0.452 | 0.103 | -2.314 | 0.60 |
| 460 | 0.271 | -0.005 | -0.723 | 0.80 | 695 | 0.191 | 0.466 | -2.481 | 0.67 |
| 465 | 0.253 | 0.048 | -0.795 | 0.81 | 700 | 0.243 | 0.310 | -2.412 | 0.62 |
| 470 | 0.267 | 0.014 | -0.815 | 0.80 | | | | | |


**Table B**. Parameters of the model of light absorption by CDOM (14) for the wavelength
range 240 - 700 nm, shown for intervals of 5 nm

| wave-length [nm] | $M$ [m$^5$ mg$^{-2}$] | $N$ [m$^2$ mg$^{-1}$] | $O$ [m$^{-1}$] | $R^2$ | wave-length [nm] | $M$ [m$^5$ mg$^{-2}$] | $N$ [m$^2$ mg$^{-1}$] | $O$ [m$^{-1}$] | $R^2$ |
|---|---|---|---|---|---|---|---|---|---|
| *1* | *2* | *3* | *4* | *5* | *1* | *2* | *3* | *4* | *5* |
| 240 | 0.337 | 0.444 | 1.360 | 0.92 | 475 | -0.300 | 1.184 | -0.572 | 0.95 |
| 245 | 0.356 | 0.445 | 1.323 | 0.94 | 480 | -0.195 | 1.129 | -0.613 | 0.95 |
| 250 | 0.369 | 0.450 | 1.294 | 0.95 | 485 | -0.211 | 1.159 | -0.657 | 0.95 |
| 255 | 0.372 | 0.455 | 1.269 | 0.95 | 490 | -0.217 | 1.147 | -0.682 | 0.93 |
| 260 | 0.375 | 0.463 | 1.243 | 0.96 | 495 | -0.226 | 1.163 | -0.720 | 0.93 |
| 265 | 0.376 | 0.474 | 1.213 | 0.96 | 500 | -0.218 | 1.163 | -0.756 | 0.92 |
| 270 | 0.370 | 0.490 | 1.177 | 0.96 | 505 | -0.176 | 1.138 | -0.787 | 0.92 |
| 275 | 0.363 | 0.511 | 1.136 | 0.96 | 510 | -0.187 | 1.150 | -0.823 | 0.90 |
| 280 | 0.355 | 0.535 | 1.091 | 0.96 | 515 | -0.206 | 1.183 | -0.867 | 0.89 |
| 285 | 0.348 | 0.562 | 1.042 | 0.96 | 520 | -0.188 | 1.174 | -0.901 | 0.88 |
| 290 | 0.340 | 0.596 | 0.988 | 0.97 | 525 | -0.140 | 1.137 | -0.929 | 0.87 |
| 295 | 0.332 | 0.633 | 0.930 | 0.97 | 530 | -0.139 | 1.149 | -0.969 | 0.88 |
| 300 | 0.317 | 0.672 | 0.873 | 0.97 | 535 | -0.182 | 1.186 | -1.005 | 0.86 |
| 305 | 0.300 | 0.709 | 0.819 | 0.97 | 540 | -0.148 | 1.158 | -1.033 | 0.86 |
| 310 | 0.283 | 0.743 | 0.767 | 0.98 | 545 | -0.197 | 1.215 | -1.082 | 0.83 |
| 315 | 0.265 | 0.771 | 0.718 | 0.98 | 550 | -0.092 | 1.150 | -1.116 | 0.82 |
| 320 | 0.247 | 0.794 | 0.673 | 0.98 | 555 | -0.025 | 1.119 | -1.155 | 0.79 |
| 325 | 0.229 | 0.813 | 0.628 | 0.98 | 560 | -0.097 | 1.192 | -1.204 | 0.77 |
| 330 | 0.212 | 0.833 | 0.584 | 0.98 | 565 | -0.157 | 1.195 | -1.217 | 0.78 |
| 335 | 0.195 | 0.851 | 0.541 | 0.98 | 570 | -0.126 | 1.174 | -1.243 | 0.76 |
| 340 | 0.185 | 0.865 | 0.497 | 0.99 | 575 | -0.081 | 1.154 | -1.282 | 0.73 |
| 345 | 0.174 | 0.880 | 0.454 | 0.99 | 580 | 0.036 | 1.130 | -1.355 | 0.74 |
| 350 | 0.167 | 0.890 | 0.411 | 0.99 | 585 | 0.187 | 1.101 | -1.434 | 0.74 |
| 355 | 0.154 | 0.902 | 0.370 | 0.99 | 590 | 0.227 | 1.022 | -1.444 | 0.70 |





| | | | | | | | | |
|---|---|---|---|---|---|---|---|---|
| **360** | 0.139 | 0.913 | 0.328 | 0.99 | **595** | 0.267 | 1.075 | -1.543 | 0.70 |
| **365** | 0.119 | 0.928 | 0.286 | 0.99 | **600** | 0.420 | 1.009 | -1.601 | 0.68 |
| **370** | 0.089 | 0.950 | 0.244 | 0.99 | **605** | 0.774 | 0.876 | -1.742 | 0.59 |
| **375** | 0.089 | 0.955 | 0.200 | 1.00 | **610** | 0.771 | 0.937 | -1.804 | 0.61 |
| **380** | 0.073 | 0.965 | 0.157 | 1.00 | **615** | 0.719 | 1.020 | -1.873 | 0.60 |
| **385** | 0.050 | 0.979 | 0.115 | 1.00 | **620** | 0.656 | 0.924 | -1.827 | 0.54 |
| **390** | 0.030 | 0.990 | 0.076 | 1.00 | **625** | 0.853 | 0.918 | -1.969 | 0.55 |
| **395** | 0.014 | 1.001 | 0.035 | 1.00 | **630** | 1.122 | 0.784 | -2.016 | 0.55 |
| **400** | 0.000 | 1.000 | 0.000 | 1.00 | **635** | 1.238 | 0.704 | -2.069 | 0.50 |
| **405** | -0.029 | 1.015 | -0.038 | 1.00 | **640** | 1.078 | 0.787 | -2.061 | 0.50 |
| **410** | -0.046 | 1.021 | -0.075 | 1.00 | **645** | 1.293 | 0.784 | -2.060 | 0.54 |
| **415** | -0.063 | 1.033 | -0.115 | 1.00 | **650** | 1.090 | 0.999 | -2.088 | 0.61 |
| **420** | -0.092 | 1.042 | -0.151 | 1.00 | **655** | 0.620 | 1.229 | -1.952 | 0.68 |
| **425** | -0.122 | 1.060 | -0.190 | 0.99 | **660** | 0.130 | 1.655 | -2.029 | 0.71 |
| **430** | -0.123 | 1.059 | -0.228 | 0.99 | **665** | -0.868 | 2.149 | -1.893 | 0.76 |
| **435** | -0.125 | 1.063 | -0.269 | 0.99 | **670** | 0.075 | 1.468 | -1.922 | 0.67 |
| **440** | -0.210 | 1.111 | -0.307 | 0.98 | **675** | -0.590 | 1.782 | -1.839 | 0.70 |
| **445** | -0.221 | 1.118 | -0.346 | 0.98 | **680** | 0.268 | 1.233 | -1.910 | 0.61 |
| **450** | -0.297 | 1.161 | -0.382 | 0.97 | **685** | -0.316 | 1.508 | -1.839 | 0.65 |
| **455** | -0.312 | 1.171 | -0.419 | 0.96 | **690** | 0.117 | 1.321 | -1.951 | 0.59 |
| **460** | -0.314 | 1.177 | -0.458 | 0.96 | **695** | -0.832 | 1.847 | -1.843 | 0.68 |
| **465** | -0.275 | 1.169 | -0.503 | 0.96 | **700** | -0.453 | 1.610 | -1.882 | 0.67 |
| **470** | -0.302 | 1.190 | -0.540 | 0.95 | | | | | |
