# Peer review of "Parameterization of the light absorption properties of chromophoric dissolved organic matter in the Baltic Sea and Pomeranian Lakes"

_Ocean Science, 2016_

## Referee Comment (RC1) · Anonymous Referee #1 · 15 Jun 2016

General comments

The paper presents very interesting work. Obviously, a lot of careful work has gone into this study and the assessment of model performance is detailed and thorough.

It does, however, not become clear what the motivation for of this work is. What are potential applications for each of the presented models and where is the advantage over previously published work? What progress has been made?

Specific comments

Why was a linear function fitted to the data in Figure 2 rather than a curve which would appear to represent the entire data range better?

[Figure]

Slope values for the data set presented seem fairly high. What about the quality of the data used to establish the models developed here: is there a dependency of slope values on concentrations which is caused by artefacts due to limited data quality (the use of a short pathlength in combination with relatively low CDOM concentrations)? Low coefficients of determination for the calculation of slope values point towards issues here.

Direct comparison of the different models (presented here and previously published) might be easier if values were presented in separate tables for every statistic metric rather than each model. Similarly, Figure 10 could be re-arranged, so that each panel shows the outputs of all models for a single chlorophyll concentration which would enable a more direct comparison.

Page 7: It would be helpful to add a short description and purpose of the different statistical metrics.

The structure, especially of the Discussion section (e.g. paragraph ll. 537ff.), should be revised as it is difficult to follow the argumentation at times. The Discussion contains paragraphs better suited in the Introduction and Results sections.

Technical comments

The language needs to be tidied up thoroughly prior to publication. It distracts from the content.

The symbols for CDOM absorption coefficients and abbreviation for chlorophyll a concentration are used inconsistently throughout the manuscript.

Lines 177 – 180: Add reference for protocol used in this work.

Line 188/ Eq. 4: Specify at which wavelength chlorophyll specific absorption coefficients calculated.

Line 198: The term 'standard deviation' is slightly mis-leading in this context as Eq. 5b

is used as descriptor of the overall error rather than variability in the data.

Line 203: Move symbol definitions to the top of the paragraph, i.e. line 197.

Line 264: How are relative RMSE values calculated? If a parameter has a logarithmic distribution, simply dividing the RMSE by the mean value creates a potential bias.

Line 452: 'uncertainty level' - Which statistical metrics does this refer to?

Line 519ff: This paragraph contains multiple subjective assessments of model performances. It would be helpful to add numbers to support the statements made.

---

## Referee Comment (RC2) · Anonymous Referee #2 · 16 Jun 2016

General comments

An interesting work in which a lot of effort for sampling, analysis and modeling has gone. The motivation for this work is not quite clear. Advantages should be more highlighted, combined with future prospects for its application.

Specific comments

Some structures should be revised, especially the discussion. Argumentation is often difficult to follow. Short and concise sentences might be easier. A little bit mixed up with other chapters, especially results.

Fig 10 could be arranged according to chlorophyll a concentrations.

[Figure]

To clarify and support statements add numbers (i), (ii), especially in the discussion section.

Technical comments

The language should be revised prior to publication.

Sentences are often too long, resulting in confusion.

Formulas should be consistent; the same goes for the description of existing parameters.

Symbols and abbreviation are used inconsistently in manuscript, e.g. chl a, CDOM. This should be revised prior to publication.

Lines 175ff: Add reference.

Line 182ff: Eq 4: At which wavelength was it calculated?

Line 203-212: not consistent with others, try to rearrange.

―――――――――――――――

---

## Author Comment (AC1) · 6 Jul 2016

General comments

The paper presents very interesting work. Obviously, a lot of careful work has gone into this study and the assessment of model performance is detailed and thorough. It does, however, not become clear what the motivation for of this work is. What are potential applications for each of the presented models and where is the advantage over previously published work? What progress has been made?

Reply: We would like to thank Reviewer 1 for appreciation of our work. We will make

effort to explain our motivation and implication of our research and proposed model in the broad context of the possible application in remote sensing, biogeochemistry and carbon cycle studies in enclosed marine basins and estuaries and fresh water lakes. The Reviewer #2 has similar remark therefore we have added a short paragraph in Introduction that fit our research in the broader aspects of applied environmental studies. Proposed new paragraph and references is included below:

"The CDOM is very reliable predictor of the dissolved organic carbon concentration in fresh and estuarine waters (Brezonik et al., 2015; Kutser et al., 2015; Tomig et al., 2016). The new ocean color operational satellite missions like the Sentinel-3 OLCI mission and pace sensors of the European Earth Observation Copernicus program and the VIIRS sensors of the US Joint Polar Satellite System program offered the medium ground resolution (in order of 250 m), which would be suitable for remote sensing observation of inland water bodies (Palmer et al., 2015; Kwiatkowska, et al, 2016). The optical properties of CDOM abundant in fresh and estuarine water at high concentration of CDOM usually shift the spectral maximum of the water transparency to solar radiation and water leaving radiance toward the longer wavelength (Darecki et al., 2003; Morel and Gentili, 2009). In extreme cases, in humic boreal lakes, the CDOM reduces the water leaving radiance intensity in the visible spectrum almost to null (Ficek et al., 2011; Ficek et al., 2012; Ylöstalo et al., 2014). To minimize this effect, the remote sensing algorithm for retrievals of the bio-optical and biogeochemical variables in optically complex waters were based on spectral bands combinations at longer wavelengths where CDOM absorption is low (e.g. Ficek et al., 2011). Therefore there is need for development of models that would enable to reconstruct the complete CDOM absorption spectrum. The detailed spectral information of CDOM absorption is needed for example to calculate the spectral indices related to molecular weight, degree of photochemical transformation (Helms, et al., 2008) or aromaticity (Weishaar et al., 2003). "

The references list has been updated with those cited in this paragraph.
Specific comments

1. Why was a linear function fitted to the data in Figure 2 Data presented on Figure 2 were showed in the semi – logarithmic scale (the aCDOM(400) on X-axis is shown in logarithmic scale, the spectral slope S is shown in linear scale). We have used the logarithmic function (equations 10, 11 and 12), to approximate relationship between aCDOM(400) and S, and therefore graphical representation of logarithmic function in a semi-logarithmic scale, is a straight line. 2. Slope values for the data set presented seem fairly high. What about the quality of the data used to establish the models developed here: is there a dependency of slope values on concentrations which is caused by artifacts due to limited data quality (the use of a short pathlength in combination with relatively low CDOM concentrations)? Low coefficients of determination for the calculation of slope values point towards issues here.

We disagree with the reviewer comment. The Baltic Sea CDOM absorption data were analyzed twice with focus of on the spectral slope values and its dependency with CDOM absorption coefficient values. The fist study published by Kowalczuk et al., (2206) presented the differences between spectral slope values calculated with different methods (linear vs. non-linear) and different spectral range used for slope calculations. We have proved in that paper that non-linear fitting methods returns higher slope values compared to linear fit log-transformed absorption data, and that the broader spectral range the smaller uncertainty is slope values would be achieved. The averaged spectral slope value $S_{300-600}$ (calculated with use of non-linear fitting method) presented in the paper by Kowalczuk et al., (2006) was 0.02334 nm-1 (n = 1610, C.V. = 12%) The second study, published by Kowalczuk et al., (2015), presented most complete to date statistical distribution of the spectral slope values in the function of salinity in the Baltic Sea. This statistical distribution has been derived upon 3636 measured aCDOM(ïĄň) spectra and the spectral slope was calculated with use the same Matlab code and in the same spectral range as we used in the current submission Presented variability range of the spectral slope S, contained within 0.015 to 0.030, plus few point

over the value of 0.030. We have also characterized the CDOM optical properties in end members: in the inflowing riverine fresh waters and in marine open Baltic Sea waters. The statistical description CDOM optical properties in open Baltic Sea waters presented in the paper by Kowalczuk et al., (2015) were as follow: salinity at the surface: $7.381 \pm 0.209$, aCDOM(350) = $1.617 \pm 0.233$ m-1, and S300-600 = $0.0232 \pm 0.0015$ nm-1, (n = 673). The fresh water end member was characterized by following average and standard deviation values: salinity = $0.918 \pm 0.546$; aCDOM(350) = $8.705 \pm 2.842$ m-1 and spectral slope coefficient S300-600 = $0.0185 \pm 0.0008$ nm-1, respectively (n = 30). The Baltic Sea data set used in the current submission were a subset of the data described by Kowalczuk et al. (2015). The same method for spectral slope calculation has been applied in to process the aCDOM(ïĄň) in lakes. In the current manuscript we presented the spectral slope variability range within 0.007 up to 0.30, both in lakes and Baltic Sea, (see, Table 2). The CDOM absorption and spectral slope variability and averaged values were very close to those already reported by Kowalczuk et al., 2006 and 2015. The lower spectral slope values were observed in lake waters, which agrees with current the knowledge about the spectral properties of CDOM absorption (CDOM absorption in fresh water is larger and absorption spectra flatter). The observed inverse dependence of the spectral slope with increasing CDOM absorption has been explained in details in paper by Stedmon and Markager (2003) and explored further in the paper by Kowalczuk et al. (2006).

We were very conservative in while performing aCDOM(ïĄň) data base re-analysis and only those spectral slope values were used in Kowalczuk et al. (2015) paper that were fitted with R2 at least 0.99. The re-analysis of CDOM absorption data based presented in paper by Kowalczuk et al., (2015), contained CDOM spectra measured with different brands of research grade spectrophotometers and different pathlengths used in measurements. We did not observed any statistical difference related to subset of data measured with different apparatus or different pathlenghts. We can assure that 5 cm cuvette used in CDOM absorption measurement in open Baltic Sea water gave similar results as CDOM absorption spectra measured with use of 10 cm cuvettes. We
quite confident in quality of our data and we do not see any issue related to low quality of data.

3. Direct comparison of the different models (presented here and previously published) might be easier if values were presented in separate tables for every statistic metric rather than each model. Similarly, Figure 10 could be re-arranged, so that each panel shows the outputs of all models for a single chlorophyll concentration which would enable a more direct comparison.

Reply: The figure 10 has been re-arranged according to reviewer suggestions.

4. Page 7: It would be helpful to add a short description and purpose of the different statistical metrics. Reply: The following paragraph has been added to explain statistical metrics used in uncertainty analysis.

Linear metrics are represented by relative mean error and standard deviation were used to measure dispersion of results and asses the modes uncertainty. The relative mean error (Eq. 5a) is the average of all relative deviations between measured and calculated values and it quantified the systematic error. Standard deviation (Eq. 5b) is the dispersion around the average error due to random errors and it quantified the statistical error. Logarithmic metrics were used to better describe the uncertainty in the data ste varying in the range of several orders of magnitude. The standard error factor described how many times the error is deviated from the average value. .

5. The structure, especially of the Discussion section (e.g. paragraph ll. 537.), should be revised as it is difficult to follow the argumentation at times. The Discussion contains paragraphs better suited in the Introduction and Results sections.

The revised manuscript structure will thoroughly corrected in terms of used argumentation and clarity. The whole manuscript will be edited to clarify the English usage, grammar and style.

Technical comments

The language needs to be tidied up thoroughly prior to publication. It distracts from the content.

Reply: We will send the revised manuscript to a English editor prior its submission to a journal editor office.

The symbols for CDOM absorption coefficients and abbreviation for chlorophyll a concentration are used inconsistently throughout the manuscript.

Reply: It has been amended.

Lines 177 – 180: Add reference for protocol used in this work.

Reply: It has been amended.

Line 188/ Eq. 4: Specify at which wavelength chlorophyll specific absorption coefficients calculated.

Reply: It has been amended.

Line 198: The term 'standard deviation' is slightly mis-leading in this context as Eq. 5b is used as descriptor of the overall error rather than variability in the data.

Line 203: Move symbol definitions to the top of the paragraph, i.e. line 197.

Reply: It has been amended.

Line 264: How are relative RMSE values calculated? If a parameter has a logarithmic distribution, simply dividing the RMSE by the mean value creates a potential bias.

Reply: All optical parameters values were presented in logarithmic scale, because in this way the relationship between these parameters (which varies with respect to more than two or three orders of magnitude) are more visible. The linear metric were applied to untransformed values of optical and bio-optical parameters. Due to broad range of variability (spanning up to three orders of magnitude) we additionally used the logarithmic metric to reduce to bias due to occurrence of very high values in the data

set, that could impact the linear metrics calculations.

Line 452: 'uncertainty level' - Which statistical metrics does this refer to?

Reply: "Uncertainty level" in this line is refer to arithmetic metric.

Line 519: This paragraph contains multiple subjective assessments of model performances. It would be helpful to add numbers to support the statements made.

Reply: We will revise the Discussion section to make our statements clear, and to assess the model performances on objective arguments.

---

## Author Comment (AC2) · 7 Jul 2016

General comments

An interesting work in which a lot of effort for sampling, analysis and modeling has gone. The motivation for this work is not quite clear. Advantages should be more highlighted, combined with future prospects for its application.

Reply: We would like to thank Reviewer 2 for appreciation of our work. We will make effort to explain our motivation and implication of our research and proposed model in the broad context of the possible application in remote sensing, biogeochemistry and carbon cycle studies in enclosed marine basins and estuaries and fresh water

lakes. The Reviewer #1 has similar remark therefore we have added a short paragraph in Introduction that fit our research in the broader aspects of applied environmental studies. Proposed new paragraph and references is included below:

"The CDOM absorption coefficient is very reliable predictor of the dissolved organic carbon concentration in fresh and estuarine waters (Brezonik et al., 2015; Kutser et al., 2015; Tomig et al., 2016). The new ocean color operational satellite missions like the Sentinel-3 OLCI mission and space sensors of the European Earth Observation Copernicus program and the VIIRS sensors of the US Joint Polar Satellite System program offered the medium ground resolution (in order of 250 m), which would be suitable for remote sensing observation of inland water bodies (Palmer et al., 2015; Kwiatkowska, et al, 2016). The optical properties of CDOM, abundant in fresh and estuarine waters at high concentrations, shift the spectral maximum of the water transparency to solar radiation and water leaving radiance toward the longer wavelength (Darecki et al., 2003; Morel and Gentili, 2009). In extreme cases, in humic boreal lakes, the CDOM reduces the water leaving radiance intensity in the visible spectrum almost to null (Ficek et al., 2011; Ficek et al., 2012; Ylöstalo et al., 2014). To minimize this effect, the remote sensing algorithm for retrievals of the bio-optical and biogeochemical variables in optically complex waters were based on spectral bands combinations at longer wavelengths where CDOM absorption is low (e.g. Ficek et al., 2011). Therefore, there is a need for development of models that would enable to reconstruct the complete CDOM absorption spectrum. The detailed spectral information of CDOM absorption is required for example to calculate the spectral indices related to molecular weight, degree of photochemical transformation (Helms, et al., 2008) or aromaticity (Weishaar et al., 2003). "

The references list has been updated with those cited in this paragraph.

Specific comments

Some structures should be revised, especially the discussion. Argumentation is often

difficult to follow. Short and concise sentences might be easier. A little bit mixed up with other chapters, especially results. Reply: The revised manuscript structure will thoroughly corrected in terms of used argumentation and clarity. The whole manuscript will be edited to clarify the English usage, grammar and style. Fig 10 could be arranged according to chlorophyll a concentrations. Reply: It has been amended.

To clarify and support statements add numbers (i), (ii), especially in the discussion section.

Reply: This remark is similar to comment by Reviewer #1. We will make effort during manuscript revision to make our statements clear, and to assess the model performances on objective arguments.

Technical comments

The language should be revised prior to publication. Reply: As we already stated the revised manuscript will corrected by professional English editor.

Sentences are often too long, resulting in confusion.

Reply: The English usage, grammar and style will be corrected by professional English editor.

Formulas should be consistent; the same goes for the description of existing parameters. Symbols and abbreviation are used inconsistently in manuscript, e.g. chl a, CDOM. This should be revised prior to publication.

Reply: It has been amended.

Lines 175ff: Add reference.

Reply: It has been amended.

Line 182ff: Eq 4: At which wavelength was it calculated?

Reply: It has been amended.

Line 203-212: not consistent with others, try to rearrange. Reply: It has been amended.
* * *

---

## Author Response (AR1)

**The response to the reviewer #1 comment on manuscript by Meler at al., Ocean Sci. Discuss., doi:10.5194/os-2016-34,**

**Anonymous Referee #1**

**General comments**

The paper presents very interesting work. Obviously, a lot of careful work has gone into this study and the assessment of model performance is detailed and thorough. It does, however, not become clear what the motivation for of this work is. What are potential applications for each of the presented models and where is the advantage over previously published work? What progress has been made?

**Reply**: We would like to thank Reviewer 1 for appreciation of our work. We will make effort to explain our motivation and implication of our research and proposed model in the broad context of the possible application in remote sensing, biogeochemistry and carbon cycle studies in enclosed marine basins and estuaries and fresh water lakes. The Reviewer #2 has similar remark therefore we have added a short paragraph in Introduction that fit our research in the broader aspects of applied environmental studies. Proposed new paragraph and references is included below:

"The CDOM absorption coefficient is a very reliable predictor of the dissolved organic carbon concentration in fresh and estuarine waters (Brezonik et al., 2015; Kutser et al., 2015; Toming et al., 2016), and therefore this optical parameter could be easily applied in various aspects of organic carbon biogeochemistry. The ocean color remote sensing offer new operational satellite missions based on medium ground resolution (of the order of 250 m) sensors, like the European Earth Observation Copernicus program Sentinel-3 OLCI mission, and the US Joint Polar Satellite System program VIIRS sensors. These radiometers are particularly suitable for remote sensing observations of inland water bodies and estuaries (Palmer et al., 2015; Kwiatkowska et al., 2016). The optical properties of CDOM, abundant in fresh and estuarine waters at high concentrations, shift the spectral maximum of the water transparency to solar radiation and water leaving radiance towards the longer wavelengths (Darecki et al., 2003; Morel and Gentili, 2009). In extreme cases, in humic boreal lakes, CDOM reduces the waterleaving radiance intensity in the visible spectrum almost to zero (Ficek et al., 2011; Ficek et al., 2012; Ylöstalo et al., 2014). To minimize this effect, the remote sensing algorithm for retrieving bio-optical and biogeochemical variables in optically complex waters has been based on spectral band combinations at longer wavelengths where CDOM absorption is low (e.g. Ficek et al., 2011). Therefore, models need to be developed that enable the complete CDOM absorption spectrum to be reconstructed. Detailed spectral information of CDOM absorption is required, for example, to calculate the spectral indices related to molecular weight, degree of photochemical transformation (Helms et al., 2008) or aromaticity (Weishaar et al., 2003). "

The references list has been updated with those cited in this paragraph.

Specific comments

1. Why was a linear function fitted to the data in Figure 2

Data presented on Figure 2 were showed in the semi – logarithmic scale (the  $a_{\text{CDOM}}(400)$  on X-axis is shown in logarithmic scale, the spectral slope *S* is shown in linear scale). We have used the logarithmic function (equations 10, 11 and 12), to approximate relationship between  $a_{\text{CDOM}}(400)$  and S, and therefore graphical representation of logarithmic function in a semi-logarithmic scale, is a straight line.

2. Slope values for the data set presented seem fairly high. What about the quality of the data used to establish the models developed here: is there a dependency of slope values on concentrations which is caused by artifacts due to limited data quality (the use of a short pathlength in combination with relatively low CDOM concentrations)? Low coefficients of determination for the calculation of slope values point towards issues here.

**We disagree with the reviewer comment.**

The Baltic Sea CDOM absorption data were analyzed twice with focus of on the spectral slope values and its dependency with CDOM absorption coefficient values.

The fist study published by Kowalczuk et al., (2206) presented the differences between spectral slope values calculated with different methods (linear vs. non-linear) and different spectral range used for slope calculations. We have proved in that paper that non-linear fitting methods returns higher slope values compared to linear fit log-transformed absorption data, and that the broader spectral range the smaller uncertainty is slope values would be achieved. The averaged spectral slope value  $S_{300-600}$  (calculated with use of non-linear fitting method) presented in the paper by Kowalczuk et al., (2006) was 0.02334 nm-1 (n = 1610, C.V. = 12%)

The second study, published by Kowalczuk et al., (2015), presented most complete to date statistical distribution of the spectral slope values in the function of salinity in the Baltic Sea. This statistical distribution has been derived upon 3636 measured  $a_{\text{CDOM}}(\lambda)$  spectra and the spectral slope was calculated with use the same Matlab code and in the same spectral range as we used in the current submission Presented variability range of the spectral slope S, contained within 0.015 to 0.030, plus few point over the value of 0.030. We have also characterized the CDOM optical properties in end members: in the inflowing riverine fresh waters and in marine open Baltic Sea waters. The statistical description CDOM optical properties in open Baltic Sea waters presented in the paper by Kowalczuk et al., (2015) were as follow: salinity at the surface: 7.381  $\pm$  0.209,  $a_{CDOM}(350) = 1.617 \pm 0.233 \text{ m}^{-1}$ , and  $S_{300-600} = 0.0232 \pm 0.0015$  nm-1, (n = 673). The fresh water end member was characterized by following average and standard deviation values: salinity =  $0.918 \pm 0.546$ ;  $a_{\text{CDOM}}(350) = 8.705 \pm 2.842 \text{ m}^{-1}$  and spectral slope coefficient  $S_{300-600} = 0.0185 \pm 0.0008 \text{ nm}^{-1}$ , salinity =  $0.918 \pm 0.546$ ; respectively (n = 30). The Baltic Sea data set used in the current submission were a subset of the data described by Kowalczuk et al. (2015). The same method for spectral slope calculation has been applied in to process the  $a_{CDOM}(\lambda)$  in lakes. In the current manuscript we presented the spectral slope variability range within 0.007 up to 0.30, both in lakes and Baltic Sea, (see, Table 2). The CDOM absorption and spectral slope variability and averaged values were very close to those already reported by Kowalczuk et al., 2006 and 2015. The lower spectral slope values were observed in lake waters, which agrees with current the knowledge about the spectral properties of CDOM absorption (CDOM absorption in fresh water is larger and absorption spectra flatter). The observed inverse dependence of the spectral slope with increasing CDOM absorption has been explained in details in paper by Stedmon and Markager (2003) and explored further in the paper by Kowalczuk et al. (2006).

We were very conservative in while performing  $a_{\text{CDOM}}(\lambda)$  data base re-analysis and only those spectral slope values were used in Kowalczuk et al. (2015) paper that were fitted with  $R^2$  at

least 0.99. The re-analysis of CDOM absorption data based presented in paper by Kowalczuk et al., (2015), contained CDOM spectra measured with different brands of research grade spectrophotometers and different pathlengths used in measurements. We did not observed any statistical difference related to subset of data measured with different apparatus or different pathlengths. We can assure that 5 cm cuvette used in CDOM absorption measurement in open Baltic Sea water gave similar results as CDOM absorption spectra measured with use of 10 cm cuvettes. We quite confident in quality of our data and we do not see any issue related to low quality of data.

3. Direct comparison of the different models (presented here and previously published) might be easier if values were presented in separate tables for every statistic metric rather than each model. Similarly, Figure 10 could be re-arranged, so that each panel shows the outputs of all models for a single chlorophyll concentration which would enable a more direct comparison.

Reply: The figure 10 has been re-arranged according to reviewer suggestions.

4. Page 7: It would be helpful to add a short description and purpose of the different statistical metrics.

Reply: The following paragraph has been added to explain statistical metrics used in uncertainty analysis.

Linear metrics are represented by relative mean error and standard deviation were used to measure dispersion of results and asses the modes uncertainty. The relative mean error (Eq. 5a) is the average of all relative deviations between measured and calculated values and it quantified the systematic error. Standard deviation (Eq. 5b) is the dispersion around the average error due to random errors and it quantified the statistical error. Logarithmic metrics were used to better describe the uncertainty in the data ste varying in the range of several orders of magnitude. The standard error factor described how many times the error is deviated from the average value.

5. The structure, especially of the Discussion section (e.g. paragraph ll. 537.), should be revised as it is difficult to follow the argumentation at times. The Discussion contains paragraphs better suited in the Introduction and Results sections.

The revised manuscript structure will thoroughly corrected in terms of used argumentation and clarity. The whole manuscript will be edited to clarify the English usage, grammar and style.

Technical comments

The language needs to be tidied up thoroughly prior to publication. It distracts from the content.

Reply: We will send the revised manuscript to a English editor prior its submission to a journal editor office.

The symbols for CDOM absorption coefficients and abbreviation for chlorophyll a concentration are used inconsistently throughout the manuscript.

Reply: It has been amended.

Lines 177 – 180: Add reference for protocol used in this work.

Reply: It has been amended.

Line 188/ Eq. 4: Specify at which wavelength chlorophyll specific absorption coefficients calculated.

Reply: It has been amended.

Line 198: The term 'standard deviation' is slightly mis-leading in this context as Eq. 5b is used as descriptor of the overall error rather than variability in the data.

Line 203: Move symbol definitions to the top of the paragraph, i.e. line 197.

Reply: It has been amended.

Line 264: How are relative RMSE values calculated? If a parameter has a logarithmic distribution, simply dividing the RMSE by the mean value creates a potential bias.

Reply:

All optical parameters values were presented in logarithmic scale, because in this way the relationship between these parameters (which varies with respect to more than two or three orders of magnitude) are more visible. The linear metric were applied to untransformed values of optical and bio-optical parameters. Due to broad range of variability (spanning up to three orders of magnitude) we additionally used the logarithmic metric to reduce to bias due to occurrence of very high values in the data set, that could impact the linear metrics calculations.

Line 452: 'uncertainty level' - Which statistical metrics does this refer to?

Reply: "Uncertainty level" in this line is refer to arithmetic metric.

Line 519: This paragraph contains multiple subjective assessments of model performances. It would be helpful to add numbers to support the statements made.

Reply: We will revise the Discussion section to make our statements clear, and to assess the model performances on objective arguments.

**The response to the reviewer #2 comment on manuscript by Meler at al., Ocean Sci. Discuss., doi:10.5194/os-2016-34,**

**Anonymous Referee #2**

**General comments**

An interesting work in which a lot of effort for sampling, analysis and modeling has gone. The motivation for this work is not quite clear. Advantages should be more highlighted, combined with future prospects for its application.

**Reply**: We would like to thank Reviewer 2 for appreciation of our work. We will make effort to explain our motivation and implication of our research and proposed model in the broad context of the possible application in remote sensing, biogeochemistry and carbon cycle studies in enclosed marine basins and estuaries and fresh water lakes. The Reviewer #1 has similar remark therefore we have added a short paragraph in Introduction that fit our research in the broader aspects of applied environmental studies. Proposed new paragraph and references is included below:

"The CDOM absorption coefficient is a very reliable predictor of the dissolved organic carbon concentration in fresh and estuarine waters (Brezonik et al., 2015; Kutser et al., 2015; Toming et al., 2016), and therefore this optical parameter could be easily applied in various aspects of organic carbon biogeochemistry. The ocean color remote sensing offer new operational satellite missions based on medium ground resolution (of the order of 250 m) sensors, like the European Earth Observation Copernicus program Sentinel-3 OLCI mission, and the US Joint Polar Satellite System program VIIRS sensors. These radiometers are particularly suitable for remote sensing observations of inland water bodies and estuaries (Palmer et al., 2015; Kwiatkowska et al., 2016). The optical properties of CDOM, abundant in fresh and estuarine waters at high concentrations, shift the spectral maximum of the water transparency to solar radiation and water leaving radiance towards the longer wavelengths (Darecki et al., 2003; Morel and Gentili, 2009). In extreme cases, in humic boreal lakes, CDOM reduces the waterleaving radiance intensity in the visible spectrum almost to zero (Ficek et al., 2011; Ficek et al., 2012; Ylöstalo et al., 2014). To minimize this effect, the remote sensing algorithm for retrieving bio-optical and biogeochemical variables in optically complex waters has been based on spectral band combinations at longer wavelengths where CDOM absorption is low (e.g. Ficek et al., 2011). Therefore, models need to be developed that enable the complete CDOM absorption spectrum to be reconstructed. Detailed spectral information of CDOM absorption is required, for example, to calculate the spectral indices related to molecular weight, degree of photochemical transformation (Helms et al., 2008) or aromaticity (Weishaar et al., 2003). "

The references list has been updated with those cited in this paragraph.

Specific comments

Some structures should be revised, especially the discussion. Argumentation is often difficult to follow. Short and concise sentences might be easier. A little bit mixed up with other chapters, especially results.

**Reply:** The revised manuscript structure will thoroughly corrected in terms of used argumentation and clarity. The whole manuscript will be edited to clarify the English usage, grammar and style.

Fig 10 could be arranged according to chlorophyll a concentrations.

Reply: It has been amended.

To clarify and support statements add numbers (i), (ii), especially in the discussion section.

Reply: This remark is similar to comment by Reviewer #1. We will make effort during manuscript revision to make our statements clear, and to assess the model performances on objective arguments.

Technical comments

The language should be revised prior to publication. Reply: As we already stated the revised manuscript will corrected by professional English editor.

Sentences are often too long, resulting in confusion.

Reply: The English usage, grammar and style will be corrected by professional English editor.

Formulas should be consistent; the same goes for the description of existing parameters. Symbols and abbreviation are used inconsistently in manuscript, e.g. chl a, CDOM. This should be revised prior to publication.

Reply: It has been amended.

Lines 175ff: Add reference.

Reply: It has been amended.

Line 182ff: Eq 4: At which wavelength was it calculated?

Reply: It has been amended.

Line 203-212: not consistent with others, try to rearrange.

Reply: It has been amended.

**matter in the Baltic Sea and Pomeranian Lakes 2 Justyna Melera\*, Piotr Kowalczuka, Mirosława Ostrowskaa, Dariusz Ficekb, Monika 3 Zabłockaa, Agnieszka Zduna 4 a Institute of Oceanology Polish Academy of Sciences, Powstańców Warszawy 55, 81-712 5 Sopot, Poland 6 b Institute of Physics, Pomeranian University of Słupsk, Bohaterów Westerplatte 64, 76-200 7 Słupsk, Poland 8 9 corresponding author: jmeler@iopan.pl 10 Keywords: Baltic Sea; Pomeranian lakes; Chromophoric Dissolved Organic Matter; three 11 alternative models of CDOM absorption; light absorption; ocean optics, 12 13 Abstract 14 This study presents three alternative models for estimatingon of the absorption properties of 15 Chromophoric Dissolved Organic Matter, $a_{CDOM}(\lambda)$ . For this analysis we used a database 16 containing 556 absorption spectra measured in 2006 - 2009 in different regions of the Baltic 17 Sea (open and coastal waters, the Gulf of Gdańsk and the Pomeranian Bay), at river mouths, 18 in the Szczecin Lagoon and also in three lakes in Pomeranian (lakes in Poland), - Lakes 19 Obłęskie, Łebsko and Chotkowskie. Observed The variability range of the CDOM absorption 20 coefficient at 400 nm, $a_{\text{CDOM}}(400)$ , contained lay within 0.15 – 8.85 m-1. The variability in 21**

Parameterization of the light absorption properties of chromophoric dissolved organic

1

 $a_{\text{CDOM}}(\lambda)$  was parameterized with respect to the three orders of magnitude-variability over 22 three orders of magnitude in the chlorophyll  $\mu$  concentration Chla (0.7 – 119 mg m-3). The 23 cChlorophyll a concentration and CDOM absorption coefficient,  $a_{CDOM}(400)$  were correlated, 24 and a statistically significant, non-linear empirical relationship between theese parameters 25 was derived ( $R^2$ =0.83). Based on On the basis of the observed co-variance between these 26 parameters, we derived two empirical mathematical models that enabled to project design the 27 CDOM absorption coefficient dynamics in natural waters and reconstruct the completed 28 updated complete CDOM absorption spectrum in the UV and visible spectral domains. The 29 input variable in the -first model was the chlorophyll a concentration, used the chlorophyll a 30 concentration as the input variable. The second model used the and in the account (400), as the 31 input variable second one it was aCDOM(400), Both models were fitted to a power function, 32

Formatted: English (U.S.) Field Code Changed Formatted: English (U.S.) Formatted: No underline, Font color: Auto, English (U.S.) Formatted: English (U.S.) Formatted: No underline, Font color: Auto, English (U.S.) Formatted: English (U.S.) Formatted: No underline, Font color: Auto, English (U.S.) Formatted: English (U.S.) Formatted: No underline, Font color: Auto, English (U.S.) Formatted ( ... Formatted ( ... Formatted (... Formatted ... Formatted ( ... Formatted (... Formatted ( ... Formatted ... Formatted ( ... Formatted **...** Formatted ( .... Formatted ( ... Formatted

and the a second\_-order polynomial function was used as the exponent. Regression coefficients for derived-these formulas were determined for wavelengths from 240 to 700 nm at 5 nm intervals-. Both approximations reflected the real shape of the absorption spectra with a low level of uncertainty. Comparison of these approximations with other models of light absorption by CDOM proved-demonstrated that-that our proposed parameterizations were better superior (bias from -1.45% to 62%, RSME from 22% to 220%) for estimationg CDOM absorption in the optically complex waters of the Baltic Sea and Pomeranian lakes. **Formatted:** No underline, Font color: Auto, English (U.S.)

Auto, English (U.S.)

| Formatted: English (U.S.)                                           |
|---------------------------------------------------------------------|
| Formatted: No underline, Font color:
Auto, English (U.S.) |
| Formatted: No underline, Font color:
Auto, English (U.S.) |
| Formatted: No underline, Font color:
Auto, English (U.S.) |
| Formatted: No underline, Font color:
Auto, English (U.S.) |
| Formatted: No underline, Font color:
Auto, English (U.S.) |
| Formatted: No underline, Font color:
Auto, English (U.S.) |

2

**40 **1. Introduction**

[revised manuscript text omitted]

| Formatted: No underline, Font color:
Auto, English (U.S.)                                                                           |
|-----------------------------------------------------------------------------------------------------------------------------------------------|
| Formatted: No underline, Font color:
Auto, English (U.S.)                                                                           |
| Formatted: No underline, Font color:
Auto, English (U.S.)                                                                           |
| Formatted: No underline, Font color:
Auto, English (U.S.)                                                                           |
| Formatted: No underline, Font color:
Auto, English (U.S.)                                                                           |
| Formatted: No underline, Font color:
Auto, English (U.S.)                                                                           |
| Formatted: No underline, Font color:
Auto, English (U.S.)                                                                           |
| Formatted: No underline, Font color:
Auto, English (U.S.)                                                                           |
| Formatted: No underline, Font color:
Auto, English (U.S.)                                                                           |
| Formatted: No underline, Font color:
Auto, English (U.S.)                                                                           |
| Formatted: No underline, Font color:
Auto, English (U.S.)                                                                                  |
|                                                                                                                                               |
| Formatted: English (U.S.)                                                                                                                     |
| Formatted: English (U.S.) Formatted: English (U.S.)                                                                                           |
| Formatted: English (U.S.)
| Formatted: English (U.S.)
| Formatted: English (U.S.)

| - | Formatted: English (U.S.) |
|---|---------------------------|
|   |                           |
| - | Formatted: English (U.S.) |
| - | Formatted: English (U.S.) |
| - | Formatted: English (U.S.) |
|   |                           |

97

CDOM plays-also plays various ecological roles in aquatic environments: even small concentrations strongly absorb UV radiation, protecting organisms from its destructive action. Higher levels of CDOM absorptions limits the amount of radiation available for photosynthesis, and consequently reducing the primary production of organic matter in that water (Górniak, 1996; Wetzel, 2001). CDOM plays an important part in the various biological processes taking place in water bodies: it can affect the species composition, number and size of plankton organisms (Arrigo and Brown, 1996; Campanelli et al., 2009), and in oligotrophic lakes can promote the growth of bacterioplankton (Moran and Hodson, 1994). Several authors have pointed out that CDOM is a potential source of reactive oxygen forms in aquatic ecosystems, which has a considerable influence on a variety of biological processes (Whitehead and de Mora, 2000; Kieber et al., 2003).

CDOM absorption decreases exponentially towards longer wavelengths and can be described by the exponential function (Jerlov, 1976, Bricaud et al., 1981,Kirk 1994):

10
$$a_{\text{CDOM}}(\lambda) = a_{\text{CDOM}}(\lambda_0)e^{-S(\lambda_0 - \lambda)}$$

1 where:  $a_{\text{CDOM}}(\lambda)$  is the light absorption coefficient for a given wavelength  $\lambda$ ,  $\lambda_0$  is the 2 reference wavelength; and S is the slope of the spectrum within a given wavelength interval.

113 The CDOM accumulates in the surface Baltic Sea waters as a combined effect of a 114 very high large inflow of fresh water from rivers, and the limited exchange of waters with the 115 North Sea and the very high productivity of in this marine basin, that sea (Kowalczuk et al., 116 2006). The Ssystematic studies over the last two decades on the optical properties in the of 117 Baltic Sea waters and its adjoining acent fresh water systems, i.e. -coastal lagoons and 118 Pomeranian lakes, have provided yielded evidence that the CDOM is the principal absorbent 119 of solar radiation and the main factor governing their optical properties (Kowalczuk 1999; 120 Kowalczuk et al., 2005a; 2006; 2010; Ficek et al., 2012; Ficek 2013).

We have performed analyses using a combined data set of optical properties of marine 121 122 and lacustrine water samples, treating the data as a single, pooled set. Other optical properties of lacustrine waters displayed a resemblance to resembled marine waters in the Baltic Sea 123 124 waters, despite the observed differences in the trophic status of theese water bodies. According to In accordance with Choiński (2007), the lakes waters were divided into ultra-125 oligotrophic, oligotrophic, mesotrophic, eutrophic, hypereutrophic and dystrophic. The 126 trophicity  $\frac{1}{2}$  was determines d by from the concentration of chlorophyll a, the water 127 transparency (determined measured by using a Secchi disk), and the concentration of biogenic 128

| $\left( \right)$ | Formatted: English (U.S.) |
|------------------|---------------------------|
| (                | Formatted: English (U.S.) |
| $\left( \right)$ | Formatted: English (U.S.) |

| Formatted: English (U.S.) |  |
|---------------------------|--|
| Formatted: English (U.S.) |  |
| Formatted: English (U.S.) |  |

(1)

factorsnutrients, e.g. nitrogen and phosphorus (Carlson, 1977; Kratzer and Brezonik, 1981).
The ranges of concentrations of chlorophyll and trophicity-defining nutrients defining
trophicity awere higher wider in lakes than in marine sea waters. In our modelling approach
we have assumed that lakes ecould ould be treated as a natural extension of coastal, lagoon
and river mouth waters.

The main objective of the present work was to derive three alternative parameterizations scenarios of the relationships between the CDOM absorption coefficient in the Baltic and Pomeranian lakes waters and physical or biogeochemical variables. The motivation for developing ment of these models was to estimate a the complete spectrum of the CDOM light absorption coefficients by using different input parameters: *i*) in the first scenario the known chlorophyll *a* concentrations in the first scenario, *ii*) in the second scenario known values of the CDOM absorption coefficient at 400 nm,  $a_{CDOM}(400)$ , in the second scenario, *iii*) and in the third scenario known values of  $a_{CDOM}(400)$  and known nonlinear relationships between CDOM absorption coefficient and the spectral slope coefficient S in the third scenario. Developed These models can be used to improve the accuracy of ocean colour remotes sensing algorithms for retrievingal of environmental variables in the Baltic Sea, adjacent estuaries river mouths, and lagoons and fresh-water lakes.

| Formatted: English (U.S.) |
|---------------------------|
|                           |
| Formatted: English (U.S.) |
| Formatted: Not Highlight  |
| Formatted: English (U.S.) |
| Formatted: Not Highlight  |
| Formatted: English (U.S.) |
|                           |

**Formatted: English (U.S.)**

**146 **2. Material and methods**

**147 2.1 Sampling area**

134

135

136 137

138

139

140

141

142 143

144

145

Water samples for determining ation of optically significant water constituents 148 concentrations were collected from August 2006 to November 2009 in the southern Baltic and 149 150 in three lakes in the Pomeranian Lake District (Poland) during the long-term observation 151 program of inherent and apparent optical properties for calibrationg and validationg of ocean colour satellite imagery products, conducted run by the Institute of Oceanology, Polish 152 Academy of Sciences, Sopot, Poland, (IOPAN). The lLocations of the 116 measuring stations, 153 154 where empirical data were gathered (a total of 413 data sets) during 16 cruises of r/v Oceania 155 on the Baltic weare shown on Figure 1, and the cruises details is are given in the Table 1. R The research cruises were organized so as to capture the dynamics of natural seasonal 156 157 variability occurring in temperate waters: i) at the end of the winter, before the onset of the spring phytoplankton bloom, when wind-driven mixing, the vertical convective thermohaline 158 circulation, reduced biological activity and reduced riverine outflow all result in clearer 159 surface waters; *ii*) in spring, when the spring phytoplankton bloom coincides with the 160

| Formatted: English (U.S.) |  |
|---------------------------|--|
| Formatted: English (U.S.) |  |

| 161 | maximum freshwater runoff from the Baltic Sea watersheddrainage basin; iii) and at the end       |
|-----|--------------------------------------------------------------------------------------------------|
| 162 | of summer, when at the peak of secondary phytoplankton blooms peak and the period of             |
| 163 | maximal thermal stratification of waters reaches its maximum extent. The geographical            |
| 164 | coverage of the samples included the Gulf of Gdańsk, the Pomeranian Bay, the Szczecin            |
| 165 | Lagoon, Polish coastal waters and the open sea (the Baltic Proper). The coastal sites in the     |
| 166 | Gulf of Gdańsk and the Pomeranian Bay are under the direct influence of two major river          |
| 167 | systems, the Vistula and the Odra, respectively, which drain the majority of Poland.             |
| 168 | Additionally, samples were collected twice a month on-at the sampling station at-on the Sopot    |
| 169 | pier (Gulf of Gdańsk), from which 66 sets of data were obtained. Field observations were also    |
| 170 | carried out from April 2006 to November 2009 on-at monthly intervals a month-(except the         |
| 171 | months when the lake surfaces of the lake wasere covered with ice) ion three Pomeranian          |
| 172 | lakes (Łebsko, Chotkowskie and Obłęskie) from which 77 data sets were obtained. Selected         |
| 173 | These lakes are enclosed water bodies with only small rivers flowing in and out of them. Lake    |
| 174 | Łebsko is a specific case, however: it is a coastal lake, and connected directly to the sea by a |
| 175 | short channelanal, Part of the land around Lake Lebsko area-immediately adjacent to the          |
| 176 | channel anal-can, on occasion, be inundated when large backflows of sea water enter the lake.    |
| 177 | The lake's water level can then rise by 50-60 cm (Chlost and Cieśliński, 2005). Such a           |
| 178 | situation obviously affects the composition and properties of the lacustrine water. Similar      |
| 179 | effects, resulting from the great variability of water properties, can be expected at the points |
| 180 | where rivers flow into lakes. The lacustrine water in these areas is thus modified by the river  |
| 181 | water.                                                                                           |

182 2.2 Samples processing

183 Discrete samples of water were taken from the surface layer of the southern Baltic and the three Pomeranian lakes with use of thea Niskin bottle. The samples for spectroscopic 184 185 measurements of CDOM light absorption were filtered twice underwent a two-step filtration 186 process:- The first filtrationonce was through acid-washed Whatman glass fibere filters (GF/F, nominal pore size 0.7 µm)-, then The water was then passed through acid-washed Sartorius 187 0.2 µm pore cellulose membrane filters to remove fine-sized particles. Spectrophotometric 188 scans of CDOM absorption spectra were performed\_done\_with use the a\_Unicam UV4-100 189 double beam spectrophotometer in the 240-700 nm spectral range; these instruments were 190 installed both-in the land base-laboratory and on board of the research ship-in the 240 700 nm 191 spectral range. The cuvette path\_length was 5 cm and the-MilliQ water was used as the 192

**Formatted: English (U.S.) Formatted: English (U.S.)**

| -{ | Formatted: English (U.S.) |
|----|---------------------------|
| -( | Formatted: English (U.S.) |
| -{ | Formatted: English (U.S.) |
|    |                           |
|    |                           |
| -{ | Formatted: English (U.S.) |
|    |                           |

| 1  | Formatted: English (U.S.)            |
|----|--------------------------------------|
| 1  | Formatted: English (U.S.), Highlight |
| -( | Formatted: English (U.S.)            |
| 1  | Formatted: English (U.S.)            |

193 reference for all measurements. The absorption coefficient  $a_{\text{CDOM}}(\lambda)$  was calculated using the 194 following equation:

| 195 | $a_{\rm CDOM}(\lambda) = 2.303 \cdot A(\lambda)/L, \tag{2}$                                                           |           | Formatted: English (U.S.)        |
|-----|-----------------------------------------------------------------------------------------------------------------------|-----------|----------------------------------|
|     |                                                                                                                       | $\square$ | Formatted: English (U.S.)        |
| 196 | where: $A(\lambda)_{\overline{i}}$ is the optical density, and L is the optical path length in meters; and the factor |           | Formatted: English (U.S.)        |
| 107 | 2 303 is the natural logarithm of 10                                                                                  | Ń         | Formatted: English (U.S.)        |
| 157 |                                                                                                                       |           | Formatted: English (U.S.)        |
| 198 | A nonlinear least squares fitting method using <del>a the</del> Trust-Region algorithm                                |           | Formatted: English (U.S.)        |
| 100 | implemented in Matlah R2000 was applied (Stedmon et al. 2000 Kowalczuk et al. 2006                                    |           | Formatted: English (U.S.)        |
| 200 | 2015) to coloridate the CDOM characterian anestrum clans coefficient S in the enestrel and                            |           |                                  |
| 200 | 2015) to calculate the CDOM absorption spectrum slope coefficient <del>,</del> 5, in the spectral range        |           |                                  |
| 201 | 300-600 nm spectral range using the following equation:                                                               |           | Formatted: English (U.S.)        |
|     | $-S(\lambda - \lambda)$                                                                                               |           | Formatted: English (U.S.)        |
| 202 | $a_{\text{CDOM}}(\lambda) = a_{\text{CDOM}}(\lambda_0)e^{-S(\lambda_0 - \lambda)} + K \tag{3}$                        |           |                                  |
| 202 | where: $\lambda_{i}$ is 350 nm and K is a background constant that allows for any baseling shift caused               |           |                                  |
| 205 | where $\tau_0$ is 550 min, and K is a background constant that anows for any baseline shift caused                    |           | Formatted: English (U.S.) |
| 204 | by residual scattering by due to fine size particle fractions, micro-air bubbles or colloidal                  |           | Formatted: English (U.S.)        |
| 205 | material present in the sample, refractive index differences between sample and the reference,                        | /         | Formatted: Not Highlight         |
| 206 | or attenuation not due to CDOM. The parameters $a_{\text{CDOM}}(350)$ , $S_{\overline{s}}$ and K were estimated       |           | Formatted: English (U.S.)        |
| 207 | simultaneously via by non-linear regression using Equation 3 (Kowalczuk et al. 2006)                                  |           | Formatted: English (U.S.)        |
| 207 | sinuitaneousiy via py non-intear regression using Equation 5 (Rowalczuk et al., 2000).                                |           | Formatted: English (U.S.)        |
| 208 | The chlorophyll a concentration was determined with useby -pigment extraction method. The                      |           | Formatted: English (U.S.)        |
| 209 | pPigments contained within the suspended particles were collected by filtration of passing the                        |           | Formatted: English (U.S.)        |
| 210 | water complex onto through 47 mm Whatman glass fiber filters (GE/E) under a low vacuum                                |           | Formatted: English (U.S.)        |
| 210 | water samples onto unodin 47-min whatman grass-riber rifers (Or/F) under a low vacuum                          |           | Formatted: English (U.S.)        |
| 211 | and extracted 24 hours in 96% ethanol at room temperature for 24 hours (Wintermans & and                              |           | Formatted: Not Highlight         |
| 212 | De Mots, 1965, Marker et al., 1980). The cehlorophyll a, Chla, concentration, Chla, was                               |           | Formatted: English (U.S.)        |
| 213 | determined spectrophotometrically with a Unicam UV4-100 spectrophotometer. (Unicam                                    |           | Formatted: English (U.S.)        |
| 215 | determined specificionical and the specific photometer (official,                                                     |           | Formatted: English (U.S.)        |
| 214 | Ltd). In this method the optical density (absorbance) of the pigment extract in ethanol at 665                        |           | Formatted: Font: Not Italic      |
| 215 | nm was corrected for the background signal in the near infrared (750 nm): $\Delta OD =$                        |           | Formatted: English (U.S.)        |
| 216 | OD(665  nm) - OD(750  nm); the absorbance was converted to the chlorophyll a concentration.                           |           | Formatted: English (U.S.)        |
| 217 | using an equation involving the volumes of filtered vector $(V)$ [dm 3 ] and athenol extract               |           | Formatted: English (U.S.)        |
| 217 | using an equation involving the volumes of intered water $(v_w)$ [uni ]; and entation extract                         |           | Formatted: English (U.S.)        |
| 218 | $(V_{EtOH})$ [cm], a 2cm cuvette path length of cuvette (l), and the chlorophyll specific        | $\vee$    | Formatted: English (U.S.)        |
| 219 | absorption coefficient of chlorophyll $\underline{a}$ in 96% ethanol $[dm^3 (g cm)^{-1}]$ (for the 665 nm)            |           | Formatted: Not Highlight         |
| 220 | [Strickland and Parsons 1972; Stramska et al., 2003]:                                                                 |           | Formatted: English (U.S.)        |
|     |                                                                                                                       |           | Formatted: English (U.S.)        |
|     |                                                                                                                       |           | Formatted: English (U.S.)        |

221

 $Chla = (10^3 \cdot \Delta OD \cdot V_{EtOH}) / (83 \cdot V_w \cdot l)^{-1}.$

(4)

7

| 222 | During the field-surveys-work, temperature and salinity profiles were measured with |
|-----|-------------------------------------------------------------------------------------|
| 223 | and SeaBird SB36 CTD probe to provide the background physical conditions during to  |
| 224 | sampling.                                                                           |

The collected data obtained were analyzed by the useing of a statistical package and 225 226 data visualization software (SigmaPlot 8.1). As the dDynamic range of variability of analyzed 227 the optical parameters values exceeded 3three orders of magnitude, therefore logarithmic transformation was applied which allowed for a better presentation of their dynamics changes 228 and to analyze statistically analyze collected the data-set accordingly. FThe following 229 230 arithmetic and logarithmic statistical metrics were used to assess the uncertainty of developed 231 the empirical relationships and models  $(X_{i,M})$  measured values;  $X_{i,C}$  estimated values (the 232 subscript *M* stands for 'measured'; subscript *C* stands for 'calculated')):

233

234

235

236

237

238

239

240

| $\alpha$ atomotord domation (atotictical arror) of $c$ (UNINE root moon callero | standard deviation | (statistical arror) | of c (DMS) | E root m |  |
|---------------------------------------------------------------------------------|--------------------|---------------------|------------|----------|--|
|---------------------------------------------------------------------------------|--------------------|---------------------|------------|----------|--|

• mean logarithmic error:
$$\langle \varepsilon \rangle_g = 10^{\left[ \langle \log(x_{i,C}/x_{i,M}) \rangle \right]} - 1$$
 (6).
• standard error factor:  $x = 10^{\sigma \log}$  (7).

statistical logarithmic errors:  $\sigma_+ = x - 1$   $\sigma_- = \frac{1}{x} - 1$ (8)

 $\langle \log(X_{i,C}/X_{i,M}) \rangle$  - mean of  $\log(X_{i,C}/X_{i,M})$ ;

 $\sigma_{\log}$  - standard deviation of the set  $\log(X_{i,C}/X_{i,M})$ .

The linear metrics are represented by the relative mean error, and the standard 241 deviation wasere used to measure the dispersion of results and assess the model's uncertainty. 242 The relative mean error (Eq. 5a) is the average of all relative deviations between measured 243 and calculated values and it-quantifiesd the systematic error. SThe standard deviation (Eq. 5b) 244 245 is the dispersion around the average error due to random errors and it-quantifiesd the statistical error. Logarithmic metrics weare used to better describe the uncertainty in the data 246

| Tormaccea. English (0.5.)                                                                                                                                                                                                                                                                                                                                                                                                                                                                                                                                                                                                                                                                                                                                                                                                                                                                                                                                                                 |
|-------------------------------------------------------------------------------------------------------------------------------------------------------------------------------------------------------------------------------------------------------------------------------------------------------------------------------------------------------------------------------------------------------------------------------------------------------------------------------------------------------------------------------------------------------------------------------------------------------------------------------------------------------------------------------------------------------------------------------------------------------------------------------------------------------------------------------------------------------------------------------------------------------------------------------------------------------------------------------------------|
| Formatted: English (U.S.)                                                                                                                                                                                                                                                                                                                                                                                                                                                                                                                                                                                                                                                                                                                                                                                                                                                                                                                                                                 |
| Formatted: Not Highlight                                                                                                                                                                                                                                                                                                                                                                                                                                                                                                                                                                                                                                                                                                                                                                                                                                                                                                                                                                  |
| Field Code Changed                                                                                                                                                                                                                                                                                                                                                                                                                                                                                                                                                                                                                                                                                                                                                                                                                                                                                                                                                                        |
| Formatted: English (U.S.), Not
Highlight                                                                                                                                                                                                                                                                                                                                                                                                                                                                                                                                                                                                                                                                                                                                                                                                                                                                                                                                               |
| Field Code Changed                                                                                                                                                                                                                                                                                                                                                                                                                                                                                                                                                                                                                                                                                                                                                                                                                                                                                                                                                                        |
| Formatted: English (U.S.), Not
Highlight                                                                                                                                                                                                                                                                                                                                                                                                                                                                                                                                                                                                                                                                                                                                                                                                                                                                                                                                               |
| Formatted: English (U.S.)                                                                                                                                                                                                                                                                                                                                                                                                                                                                                                                                                                                                                                                                                                                                                                                                                                                                                                                                                                 |
| Formatted: English (U.S.)                                                                                                                                                                                                                                                                                                                                                                                                                                                                                                                                                                                                                                                                                                                                                                                                                                                                                                                                                                 |
|                                                                                                                                                                                                                                                                                                                                                                                                                                                                                                                                                                                                                                                                                                                                                                                                                                                                                                                                                                                           |
| Formatted: English (U.S.)                                                                                                                                                                                                                                                                                                                                                                                                                                                                                                                                                                                                                                                                                                                                                                                                                                                                                                                                                                 |
| Formatted: English (U.S.) Formatted: English (U.S.)                                                                                                                                                                                                                                                                                                                                                                                                                                                                                                                                                                                                                                                                                                                                                                                                                                                                                                                                       |
| Formatted: English (U.S.)
Formatted: English (U.S.)                                                                                                                                                                                                                                                                                                                                                                                                                                                                                                                                                                                                                                                                                                                                                                                                                                                                                                       |
| Formatted: English (U.S.)
Formatted: English (U.S.)                                                                                                                                                                                                                                                                                                                                                                                                                                                                                                                                                                                                                                                                                                                                                                                                                                                                          |
| Formatted: English (U.S.)
Formatted: English (U.S.)                                                                                                                                                                                                                                                                                                                                                                                                                                                                                                                                                                                                                                                                                                                                                                                                                                             |
| Formatted: English (U.S.)
Formatted: English (U.S.)                                                                                                                                                                                                                                                                                                                                                                                                                                                                                                                                                                                                                                                                                                                                                                                                                |
| Formatted: English (U.S.)
Formatted: English (U.S.)                                                                                                                                                                                                                                                                                                                                                                                                                                                                                                                                                                                                                                                                                                                                                                                   |
| Formatted: English (U.S.)
Formatted: English (U.S.)                                                                                                                                                                                                                                                                                                                                                                                                                                                                                                                                                                                                                                                                                                                                                      |
| Formatted: English (U.S.)
Formatted: English (U.S.)                                                                                                                                                                                                                                                                                                                                                                                                                                                                                                                                                                                                                                                                                                                         |
| Formatted: English (U.S.)
Formatted: English (U.S.)                                                                                                                                                                                                                                                                                                                                                                                                                                                                                                                                                                                                                                                                                            |
| Formatted: English (U.S.)
Formatted: English (U.S.)                                                                                                                                                                                                                                                                                                                                                                                                                                                                                                                                                                                                                                                               |
| Formatted: English (U.S.)
Formatted: English (U.S.)                                                                                                                                                                                                                                                                                                                                                                                                                                                                                                                                                                                                                                  |
| Formatted: English (U.S.)
Formatted: English (U.S.)                                                                                                                                                                                                                                                                                                                                                                                                                                                                                                                                                                        |
| Formatted: English (U.S.)
Formatted: English (U.S.)                                                                                                                                                                                                                                                                                                                                                                                                                                                                                                              |
| Formatted: English (U.S.)
Formatted: English (U.S.)                                                                                                                                                                                                                                                                                                                                                                                                                                                                                 |
| Formatted: English (U.S.)
Formatted: English (U.S.)                                                                                                                                                                                                                                                                                                                                                                                                                       |
| Formatted: English (U.S.)
Formatted: English (U.S.)                                                                                                                                                                                                                                                                                                                                |
| Formatted: English (U.S.)
Formatted: English (U.S.)                                                                                                                                                                                                                                                                      |
| Formatted: English (U.S.)
Formatted: English (U.S.)                                                                                                                                                                                                            |
| Formatted: English (U.S.)
| Formatted: English (U.S.)
| Formatted: English (U.S.)

(7)

stet varying in the range of over several orders of magnitude. The standard error factor
describesd how many times the error is deviatesd from the average value.

249 **3. Results**

250 251 3.1 Variability of *analysed\_the* parameters and empirical relationship between CDOM absorption and spectral slope coefficient.

Table 2 lists the Vyariability range and average values of selected optical parameters 252 253 measured in the study area and used for formulating the empirical model; the light absorption coefficients by CDOM at two wavelengths: (375 and 400 nm);  $-a_{CDOM}(375)$  and  $a_{CDOM}(400)$ ; 254 spectral slope S, and chlorophyll a concentrations, Chla., measured in the study area and used 255 256 for formulation of empirical model have been presented in the Table 2. The minima in of the variability ranges of  $a_{\text{CDOM}}(375)$ ,  $a_{\text{CDOM}}(400)$  and Chla, were noted reached a minimum in sea 257 watersin marine waters. The minimalum values of CDOM absorption coefficients in 258 259 lacustrine waters were almost an one order of magnitude higher than in marine sea waters, indicating a significant accumulation of CDOM in fresh waters. The maximalum values of 260  $a_{\text{CDOM}}(375)$ ,  $a_{\text{CDOM}}(400)$  and *Chla* were observed recorded in fresh waters; these maximal 261 262 values were approximately two time twice as high as higher than those values of the respective parameters in marine sea waters. Consequently, the average values of the CDOM 263 absorption coefficients:  $(a_{CDOM}(375), a_{CDOM}(400))$  and chlorophyll a concentrations, were 264 higher in fresh waters compared to marine than in sea waters. -The reverse trend is observed 265 266 was reversed in the case of the CDOM absorption spectrum slope coefficient,  $S_{\tau}$  and its variability range: both of the minimal maximum and maximal minimum spectral slopes, 267 values were lower in the lakes than those observed in the marine sea waters. The average 268 269 value of the spectral slope coefficient was higher in marine sea waters than in lake waters. These two data sets, measured in the the Baltic Sea waters and Pomeranian lakes, were 270 statistically significantly different, as indicated by the results of simple analysis of variance: 271  $(p-p = -3.4, -10^{-38})$ . However, their variability ranges were such, that the data from the two 272 273 different aquatic environments were overlapped, ing creating a coherent data set, that could be analyszed togetherjointly. Our principle assumption for when the derivation of deriving the 274 CDOM absorption model was that, the optical properties of lacustrine waters could be treated 275 276 as if they were an extension of estuarine and marine sea waters.

The spectral slope coefficient was inversely and non-linearly related with to the
CDOM absorption coefficient. The highly absorbing samples were spectrally flatter

| l | Formatted: English (U.S.) |
|---|---------------------------|
|   | Formatted: English (U.S.) |
| ĺ | Formatted: English (U.S.) |
|   |                           |
| ſ | Formatted: English (U.S.) |

Formatted

( ... )

Formatted

| 279 | (characteriszed by a lower S value). Different functional types were used to model this                                                                                                                                                                                                                                                                                                                                                                                                                                                                                                                                                                                                                                                                                                                                                                                                                                                                                                                                                                                                                                                                                                                                                                                                                                                                                                                                                                                                                                                                                                                                                                                                                                                                                                                                                                                                                                                                                                                                                                                                                                                                                                                                                                                                                                                                                                                                                                                                                                                                                                                                                                                                                                                                                                                                                                                                                                                                                     |    | Formatted                 |          |
|-----|------------------------------------------------------------------------------------------------------------------------------------------------------------------------------------------------------------------------------------------------------------------------------------------------------------------------------------------------------------------------------------------------------------------------------------------------------------------------------------------------------------------------------------------------------------------------------------------------------------------------------------------------------------------------------------------------------------------------------------------------------------------------------------------------------------------------------------------------------------------------------------------------------------------------------------------------------------------------------------------------------------------------------------------------------------------------------------------------------------------------------------------------------------------------------------------------------------------------------------------------------------------------------------------------------------------------------------------------------------------------------------------------------------------------------------------------------------------------------------------------------------------------------------------------------------------------------------------------------------------------------------------------------------------------------------------------------------------------------------------------------------------------------------------------------------------------------------------------------------------------------------------------------------------------------------------------------------------------------------------------------------------------------------------------------------------------------------------------------------------------------------------------------------------------------------------------------------------------------------------------------------------------------------------------------------------------------------------------------------------------------------------------------------------------------------------------------------------------------------------------------------------------------------------------------------------------------------------------------------------------------------------------------------------------------------------------------------------------------------------------------------------------------------------------------------------------------------------------------------------------------------------------------------------------------------------------------------------------------------|----|---------------------------|----------|
| 280 | relationships: hHyperbolic (Stedmon and Markager, 2001, Kowalczuk et al., 2006), and or                                                                                                                                                                                                                                                                                                                                                                                                                                                                                                                                                                                                                                                                                                                                                                                                                                                                                                                                                                                                                                                                                                                                                                                                                                                                                                                                                                                                                                                                                                                                                                                                                                                                                                                                                                                                                                                                                                                                                                                                                                                                                                                                                                                                                                                                                                                                                                                                                                                                                                                                                                                                                                                                                                                                                                                                                                                                                            |    |                           |          |
| 281 | logarithmic (Kowalczuk et al., 2005b) functional types were used to model this relationship.                                                                                                                                                                                                                                                                                                                                                                                                                                                                                                                                                                                                                                                                                                                                                                                                                                                                                                                                                                                                                                                                                                                                                                                                                                                                                                                                                                                                                                                                                                                                                                                                                                                                                                                                                                                                                                                                                                                                                                                                                                                                                                                                                                                                                                                                                                                                                                                                                                                                                                                                                                                                                                                                                                                                                                                                                                                                                       |    |                           |          |
| 282 | For consistency with Kowalczuk (2001), we have used the log-linear fit to describe the                                                                                                                                                                                                                                                                                                                                                                                                                                                                                                                                                                                                                                                                                                                                                                                                                                                                                                                                                                                                                                                                                                                                                                                                                                                                                                                                                                                                                                                                                                                                                                                                                                                                                                                                                                                                                                                                                                                                                                                                                                                                                                                                                                                                                                                                                                                                                                                                                                                                                                                                                                                                                                                                                                                                                                                                                                                                                             |    |                           |          |
| 283 | relationship between $a_{\text{CDOM}}(400)$ and S. The distribution of the spectral slope in the as a                                                                                                                                                                                                                                                                                                                                                                                                                                                                                                                                                                                                                                                                                                                                                                                                                                                                                                                                                                                                                                                                                                                                                                                                                                                                                                                                                                                                                                                                                                                                                                                                                                                                                                                                                                                                                                                                                                                                                                                                                                                                                                                                                                                                                                                                                                                                                                                                                                                                                                                                                                                                                                                                                                                                                                                                                                                                              |    |                           |          |
| 284 | function of the CDOM absorption coefficient in the Baltic Sea (black dots) and Pomeranian                                                                                                                                                                                                                                                                                                                                                                                                                                                                                                                                                                                                                                                                                                                                                                                                                                                                                                                                                                                                                                                                                                                                                                                                                                                                                                                                                                                                                                                                                                                                                                                                                                                                                                                                                                                                                                                                                                                                                                                                                                                                                                                                                                                                                                                                                                                                                                                                                                                                                                                                                                                                                                                                                                                                                                                                                                                                                          | // |                           |          |
| 285 | lakes (green dots) <del>has been presented is shown o</del> in <del>the F</del> igure 2a. The black line re presents the                                                                                                                                                                                                                                                                                                                                                                                                                                                                                                                                                                                                                                                                                                                                                                                                                                                                                                                                                                                                                                                                                                                                                                                                                                                                                                                                                                                                                                                                                                                                                                                                                                                                                                                                                                                                                                                                                                                                                                                                                                                                                                                                                                                                                                                                                                                                                                                                                                                                                                                                                                                                                                                                                                                                                                                                                                             | /  |                           |          |
| 286 | log-linear dependence (Equation 9) cobtained by Kowalczuk (2001), overlaiding on our data set:                                                                                                                                                                                                                                                                                                                                                                                                                                                                                                                                                                                                                                                                                                                                                                                                                                                                                                                                                                                                                                                                                                                                                                                                                                                                                                                                                                                                                                                                                                                                                                                                                                                                                                                                                                                                                                                                                                                                                                                                                                                                                                                                                                                                                                                                                                                                                                                                                                                                                                                                                                                                                                                                                                                                                                                                                                                                                     |    | Formatted: English (U.S.) |          |
|     |                                                                                                                                                                                                                                                                                                                                                                                                                                                                                                                                                                                                                                                                                                                                                                                                                                                                                                                                                                                                                                                                                                                                                                                                                                                                                                                                                                                                                                                                                                                                                                                                                                                                                                                                                                                                                                                                                                                                                                                                                                                                                                                                                                                                                                                                                                                                                                                                                                                                                                                                                                                                                                                                                                                                                                                                                                                                                                                                                                                    |    | ( -                |          |
| 287 | $S = log[1.038 a_{CDOM}(400)^{-0.022}].$ (9)                                                                                                                                                                                                                                                                                                                                                                                                                                                                                                                                                                                                                                                                                                                                                                                                                                                                                                                                                                                                                                                                                                                                                                                                                                                                                                                                                                                                                                                                                                                                                                                                                                                                                                                                                                                                                                                                                                                                                                                                                                                                                                                                                                                                                                                                                                                                                                                                                                                                                                                                                                                                                                                                                                                                                                                                                                                                                                                                       |    | Formatted                 |          |
| 288 | The old realation his worksed satisfactorily for part of the Baltic Sea data set                                                                                                                                                                                                                                                                                                                                                                                                                                                                                                                                                                                                                                                                                                                                                                                                                                                                                                                                                                                                                                                                                                                                                                                                                                                                                                                                                                                                                                                                                                                                                                                                                                                                                                                                                                                                                                                                                                                                                                                                                                                                                                                                                                                                                                                                                                                                                                                                                                                                                                                                                                                                                                                                                                                                                                                                                                                                                                   |    | Formatted                 |          |
| 289 | $(R^2 = 0.76)$ - but it does not cover a large group of CDOM absorption coefficients values larger                                                                                                                                                                                                                                                                                                                                                                                                                                                                                                                                                                                                                                                                                                                                                                                                                                                                                                                                                                                                                                                                                                                                                                                                                                                                                                                                                                                                                                                                                                                                                                                                                                                                                                                                                                                                                                                                                                                                                                                                                                                                                                                                                                                                                                                                                                                                                                                                                                                                                                                                                                                                                                                                                                                                                                                                                                                                                 |    |                           |          |
| 200 | then $> 5 \text{ m}^{-1}$ . The values of $q = -(400) > 5 \text{ m}^{-1}$ were measured in the lakes and in                                                                                                                                                                                                                                                                                                                                                                                                                                                                                                                                                                                                                                                                                                                                                                                                                                                                                                                                                                                                                                                                                                                                                                                                                                                                                                                                                                                                                                                                                                                                                                                                                                                                                                                                                                                                                                                                                                                                                                                                                                                                                                                                                                                                                                                                                                                                                                                                                                                                                                                                                                                                                                                                                                                                                                                                                                                                        |    |                           |          |
| 290 | $\frac{1}{1}$ $\frac{1}{2}$ $\frac{1}$ |    |                           |          |
| 291 | estuarine waters, as well as and in the Szczecin Lagoon and where the waters of the Vistula                                                                                                                                                                                                                                                                                                                                                                                                                                                                                                                                                                                                                                                                                                                                                                                                                                                                                                                                                                                                                                                                                                                                                                                                                                                                                                                                                                                                                                                                                                                                                                                                                                                                                                                                                                                                                                                                                                                                                                                                                                                                                                                                                                                                                                                                                                                                                                                                                                                                                                                                                                                                                                                                                                                                                                                                                                                                                        |    |                           |          |
| 292 | and Odra mouth infiguoung into the southern Baltic. We have derived a new formulea to                                                                                                                                                                                                                                                                                                                                                                                                                                                                                                                                                                                                                                                                                                                                                                                                                                                                                                                                                                                                                                                                                                                                                                                                                                                                                                                                                                                                                                                                                                                                                                                                                                                                                                                                                                                                                                                                                                                                                                                                                                                                                                                                                                                                                                                                                                                                                                                                                                                                                                                                                                                                                                                                                                                                                                                                                                                                                              |    |                           |          |
| 293 | determine the $a_{\text{CDOM}}(400)/S$ relationship that covered the whole range of the $a_{\text{CDOM}}(400)$                                                                                                                                                                                                                                                                                                                                                                                                                                                                                                                                                                                                                                                                                                                                                                                                                                                                                                                                                                                                                                                                                                                                                                                                                                                                                                                                                                                                                                                                                                                                                                                                                                                                                                                                                                                                                                                                                                                                                                                                                                                                                                                                                                                                                                                                                                                                                                                                                                                                                                                                                                                                                                                                                                                                                                                                                                                                     | // |                           |          |
| 294 | observed recorded in both the Baltic Sea and in Pomeranian lakes waters. The new formulea                                                                                                                                                                                                                                                                                                                                                                                                                                                                                                                                                                                                                                                                                                                                                                                                                                                                                                                                                                                                                                                                                                                                                                                                                                                                                                                                                                                                                                                                                                                                                                                                                                                                                                                                                                                                                                                                                                                                                                                                                                                                                                                                                                                                                                                                                                                                                                                                                                                                                                                                                                                                                                                                                                                                                                                                                                                                                          |    |                           |          |
| 295 | was is marked shown oin Figure 2-a as a red curve and is described by Equation 10-:                                                                                                                                                                                                                                                                                                                                                                                                                                                                                                                                                                                                                                                                                                                                                                                                                                                                                                                                                                                                                                                                                                                                                                                                                                                                                                                                                                                                                                                                                                                                                                                                                                                                                                                                                                                                                                                                                                                                                                                                                                                                                                                                                                                                                                                                                                                                                                                                                                                                                                                                                                                                                                                                                                                                                                                                                                                                                                |    | Formatted: English (U.S.) |          |
| 296 | $S = 0.0213 - 0.003 \ln[a_{CDOM}(400)]. \tag{10}$                                                                                                                                                                                                                                                                                                                                                                                                                                                                                                                                                                                                                                                                                                                                                                                                                                                                                                                                                                                                                                                                                                                                                                                                                                                                                                                                                                                                                                                                                                                                                                                                                                                                                                                                                                                                                                                                                                                                                                                                                                                                                                                                                                                                                                                                                                                                                                                                                                                                                                                                                                                                                                                                                                                                                                                                                                                                                                                                  |    | Formatted                 |          |
|     |                                                                                                                                                                                                                                                                                                                                                                                                                                                                                                                                                                                                                                                                                                                                                                                                                                                                                                                                                                                                                                                                                                                                                                                                                                                                                                                                                                                                                                                                                                                                                                                                                                                                                                                                                                                                                                                                                                                                                                                                                                                                                                                                                                                                                                                                                                                                                                                                                                                                                                                                                                                                                                                                                                                                                                                                                                                                                                                                                                                    |    | Formatted: English (U.S.) |          |
| 297 | The new $a_{\text{CDOM}}(400)$ /S relationship has been found is much better constrained and explainsed                                                                                                                                                                                                                                                                                                                                                                                                                                                                                                                                                                                                                                                                                                                                                                                                                                                                                                                                                                                                                                                                                                                                                                                                                                                                                                                                                                                                                                                                                                                                                                                                                                                                                                                                                                                                                                                                                                                                                                                                                                                                                                                                                                                                                                                                                                                                                                                                                                                                                                                                                                                                                                                                                                                                                                                                                                                                            | 1  | Formatted                 |          |
| 298 | much more variance ( $R^2 = 0.79$ ) with less uncertainty (RMSE = 0.1%) compared to the one                                                                                                                                                                                                                                                                                                                                                                                                                                                                                                                                                                                                                                                                                                                                                                                                                                                                                                                                                                                                                                                                                                                                                                                                                                                                                                                                                                                                                                                                                                                                                                                                                                                                                                                                                                                                                                                                                                                                                                                                                                                                                                                                                                                                                                                                                                                                                                                                                                                                                                                                                                                                                                                                                                                                                                                                                                                                                        | // |                           |          |
| 299 | presented given by Kowalczuk (2001).                                                                                                                                                                                                                                                                                                                                                                                                                                                                                                                                                                                                                                                                                                                                                                                                                                                                                                                                                                                                                                                                                                                                                                                                                                                                                                                                                                                                                                                                                                                                                                                                                                                                                                                                                                                                                                                                                                                                                                                                                                                                                                                                                                                                                                                                                                                                                                                                                                                                                                                                                                                                                                                                                                                                                                                                                                                                                                                                               |    | Formatted: English (U.S.) |          |
| 200 | Detailed analysis of the spectral slope distribution of spectral slope in the as a function                                                                                                                                                                                                                                                                                                                                                                                                                                                                                                                                                                                                                                                                                                                                                                                                                                                                                                                                                                                                                                                                                                                                                                                                                                                                                                                                                                                                                                                                                                                                                                                                                                                                                                                                                                                                                                                                                                                                                                                                                                                                                                                                                                                                                                                                                                                                                                                                                                                                                                                                                                                                                                                                                                                                                                                                                                                                                        |    | Cormottod                 |          |
| 201 | $c_{\text{f}} = (400)$ indicated that the date set could be divided in with respect to colinity into two                                                                                                                                                                                                                                                                                                                                                                                                                                                                                                                                                                                                                                                                                                                                                                                                                                                                                                                                                                                                                                                                                                                                                                                                                                                                                                                                                                                                                                                                                                                                                                                                                                                                                                                                                                                                                                                                                                                                                                                                                                                                                                                                                                                                                                                                                                                                                                                                                                                                                                                                                                                                                                                                                                                                                                                                                                                                           |    | ronnatteu                 |   |
| 202 | of $u_{\text{CDOM}}(400)$ indicated that the data set could be divided $\frac{1}{1000}$ with respect to samily into two                                                                                                                                                                                                                                                                                                                                                                                                                                                                                                                                                                                                                                                                                                                                                                                                                                                                                                                                                                                                                                                                                                                                                                                                                                                                                                                                                                                                                                                                                                                                                                                                                                                                                                                                                                                                                                                                                                                                                                                                                                                                                                                                                                                                                                                                                                                                                                                                                                                                                                                                                                                                                                                                                                                                                                                                                                                            |    |                           |          |
| 302 | subsets. samples characterized by saminty $\frac{1}{10000000000000000000000000000000000$                                                                                                                                                                                                                                                                                                                                                                                                                                                                                                                                                                                                                                                                                                                                                                                                                                                                                                                                                                                                                                                                                                                                                                                                                                                                                                                                                                                                                                                                                                                                                                                                                                                                                                                                                                                                                                                                                                                                                                                                                                                                                                                                                                                                                                                                                                                                                                                                                                                                                                                                                                                                                                                                                                                                                                                                                                                                                           |    |                           |          |
| 303 | those with samily $\frac{1}{2}$ , which include waters from fiver mouths, lakes and the                                                                                                                                                                                                                                                                                                                                                                                                                                                                                                                                                                                                                                                                                                                                                                                                                                                                                                                                                                                                                                                                                                                                                                                                                                                                                                                                                                                                                                                                                                                                                                                                                                                                                                                                                                                                                                                                                                                                                                                                                                                                                                                                                                                                                                                                                                                                                                                                                                                                                                                                                                                                                                                                                                                                                                                                                                                                                            |    |                           |          |
| 304 | Szczecin Lagoon. The relationship between $a_{\text{CDOM}}(400)$ and S derived for the respective data                                                                                                                                                                                                                                                                                                                                                                                                                                                                                                                                                                                                                                                                                                                                                                                                                                                                                                                                                                                                                                                                                                                                                                                                                                                                                                                                                                                                                                                                                                                                                                                                                                                                                                                                                                                                                                                                                                                                                                                                                                                                                                                                                                                                                                                                                                                                                                                                                                                                                                                                                                                                                                                                                                                                                                                                                                                                             |    |                           |          |
| 305 | substets weare presented ion Figure 2-b and the functional formulaes weare given by                                                                                                                                                                                                                                                                                                                                                                                                                                                                                                                                                                                                                                                                                                                                                                                                                                                                                                                                                                                                                                                                                                                                                                                                                                                                                                                                                                                                                                                                                                                                                                                                                                                                                                                                                                                                                                                                                                                                                                                                                                                                                                                                                                                                                                                                                                                                                                                                                                                                                                                                                                                                                                                                                                                                                                                                                                                                                                |    | (                         |          |
| 306 | Equations 11 (salinity $> 5$ ) and Equation 12 (salinity $< 5$ ):                                                                                                                                                                                                                                                                                                                                                                                                                                                                                                                                                                                                                                                                                                                                                                                                                                                                                                                                                                                                                                                                                                                                                                                                                                                                                                                                                                                                                                                                                                                                                                                                                                                                                                                                                                                                                                                                                                                                                                                                                                                                                                                                                                                                                                                                                                                                                                                                                                                                                                                                                                                                                                                                                                                                                                                                                                                                                                                  |    | Formatted: English (U.S.) |          |
| 307 | $S = 0.0206 - 0.004 \ln[a_{\text{CDOM}}(400)]$ (11)                                                                                                                                                                                                                                                                                                                                                                                                                                                                                                                                                                                                                                                                                                                                                                                                                                                                                                                                                                                                                                                                                                                                                                                                                                                                                                                                                                                                                                                                                                                                                                                                                                                                                                                                                                                                                                                                                                                                                                                                                                                                                                                                                                                                                                                                                                                                                                                                                                                                                                                                                                                                                                                                                                                                                                                                                                                                                                                                |    | Formatted                 |          |
|     |                                                                                                                                                                                                                                                                                                                                                                                                                                                                                                                                                                                                                                                                                                                                                                                                                                                                                                                                                                                                                                                                                                                                                                                                                                                                                                                                                                                                                                                                                                                                                                                                                                                                                                                                                                                                                                                                                                                                                                                                                                                                                                                                                                                                                                                                                                                                                                                                                                                                                                                                                                                                                                                                                                                                                                                                                                                                                                                                                                                    |    | Formatted                 |          |
| 308 | $S = 0.0196 - 0.0009 \ln[a_{\text{CDOM}}(400)].$ (12)                                                                                                                                                                                                                                                                                                                                                                                                                                                                                                                                                                                                                                                                                                                                                                                                                                                                                                                                                                                                                                                                                                                                                                                                                                                                                                                                                                                                                                                                                                                                                                                                                                                                                                                                                                                                                                                                                                                                                                                                                                                                                                                                                                                                                                                                                                                                                                                                                                                                                                                                                                                                                                                                                                                                                                                                                                                                                                                              |    | Formatted: English (U.S.) |          |
|     | 10                                                                                                                                                                                                                                                                                                                                                                                                                                                                                                                                                                                                                                                                                                                                                                                                                                                                                                                                                                                                                                                                                                                                                                                                                                                                                                                                                                                                                                                                                                                                                                                                                                                                                                                                                                                                                                                                                                                                                                                                                                                                                                                                                                                                                                                                                                                                                                                                                                                                                                                                                                                                                                                                                                                                                                                                                                                                                                                                                                                 |    |                           |          |

| 309 | The suggested Proposed approximations of the $\mu_{CDOM}(400)/S$ relationships in the two salinity    | 1   | Formatted                 |  |
|-----|-------------------------------------------------------------------------------------------------------|-----|---------------------------|--|
| 310 | ranges we have re characterised by the a higher explained variance ( $R^2 = 0.78$ for Equation 11,    |     |                           |  |
| 311 | and lower $R^2 = 0.22$ , for Equation 12), respectively. In both cases, the estimation uncertainties  | /// |                           |  |
| 312 | -y : RSME = $0.08\%$ for Equation 11, and RSME = $0.09\%$ , for Equation 12, respectively, _   | //  |                           |  |
| 313 | were lower compared to than the approximation presented by given by Equation 10.                      |     | Formatted: English (U.S.) |  |
| 314 | 3.2. A model for approximating on of the CDOM light absorption spectrum from the empirical            |     | Formatted                 |  |
| 315 | dependence on with the chlorophyll a concentration.                                                   |     | Formatted: English (U.S.) |  |
|     |                                                                                                       |     |                           |  |
| 316 | The principle bio-optical assumption on interdependencies among optically significant                 | 1   | Formatted                 |  |
| 317 | water constituents in global-the world ocean was formulated by Morel and Prieur (1977), who           |     |                           |  |
| 318 | introduced the concept of the Case 1 waters, where the variability iof those constituents wais        |     |                           |  |
| 319 | to far-a considerable extent correlated with the variability of in the phytoplankton biomass          |     |                           |  |
| 320 | expressed as chlorophyll a concentration. The Case 1 waters we were mostly open oceanic        |     |                           |  |
| 321 | waters and upwelling regions at along western continental margins. The marine basins sea              |     |                           |  |
| 322 | areas where these is assumption were was not fulfilled, we were considered treated as Case 2          |     |                           |  |
| 323 | waters; mostly semi-enclosed and shelf seas and coastal oceans, where there wewere sources            |     |                           |  |
| 324 | of riverine waters. It was assumed that changes in the magnitude of optically significant water       |     |                           |  |
| 325 | constituents in the Case 2 waters were independent. This concept was critically reassessed by         |     |                           |  |
| 326 | Siegel et al. (2005), who reanalyzed the global ocean colour imagery data set. They, and              |     |                           |  |
| 327 | demonstrated proved-that, although in open ocean-the bio-optical assumption iwas still valid          |     |                           |  |
| 328 | in the open ocean, there were significant dependences between chlorophyll $p$ and other               |     |                           |  |
| 329 | optically significant water constituents at regional scales in along oceanic continental              |     |                           |  |
| 330 | margins. Even though the CDOM was not thought to be correlated with chlorophyll a                     |     |                           |  |
| 331 | concentrations in Case 2 waters, there were examples showing that such a relationships were           |     |                           |  |
| 332 | was possible (Ferrari and Tassan, 1992; Vodacek et al., 1997). In the-Baltic waters such              |     |                           |  |
| 333 | analyses were carried out by Kowalczuk and Kaczmarek (1996) and Kowalczuk (1999).                     |     |                           |  |
| 334 | These authors demonstrated that the correlation between the concentration of chlorophyll a     |     |                           |  |
| 335 | and the CDOM absorption coefficient was observed were correlated. The positive correlation            |     |                           |  |
| 336 | between light absorption by CDOM and chlorophyll a concentration has been confirmed with       |     |                           |  |
| 337 | new data available, from both in marine sea and fresh waters. The elear clearly increasing            |     |                           |  |
| 338 | trend of increase of the CDOM absorption level with increasing phytoplankton biomass has              |     |                           |  |
| 339 | been presented on is shown in Figure 3. The dependence between $a_{\text{CDOM}}(400)$ coefficient and |     |                           |  |
| 340 | the concentration Chla obtained by Kowalczuk (2001) has been overlaind on the new,                    |     | Field Code Changed        |  |
| 341 | currently reported updated empirical data set, in Figure 3. It is evident that the                    | >   | Formatted                 |  |

| 342  | $a_{\text{CDOM}}(400)/Chla$ relationship reported by Kowalczuk is applicable to only some of the Baltic                                                                                  |           | Formatted                 |  |
|------|------------------------------------------------------------------------------------------------------------------------------------------------------------------------------------------|-----------|---------------------------|--|
| 343  | Sea data, in the chlorophyll a -concentration range $0.8 < Chla < 10$ mg m -3 . The old, previous                                                                      | /         |                           |  |
| 344  | power function relationship did not reproduced correctly the $a_{\text{CDOM}}(400)$ values for high                                                                                      |           |                           |  |
| 345  | chlorophyll a concentrations, and CDOM absorption data measured in estuaries in river                                                                                             | ///       |                           |  |
| 346  | mouths and lakes were lying lay above the model curve. We have proposed a new                                                                                                            | //        |                           |  |
| 347  | statistically significant relationship between the $a_{CDOM}(400)$ and Chla which was that is                                                                                            |           |                           |  |
| 348  | described by a second-degree polynomial ( $R^2 = 0.83$ , RMSE = 28%, n = 541, p<0.0001).                                                                                                 |           | Formatted: English (U.S.) |  |
| 3/10 | The same function has been applied to reconstruct the complete CDOM absorption                                                                                                           |           | Formatted                 |  |
| 350  | spectrum in the spectral range from 245 to 700 nm with 5 nm resolution- (Equation 13):                                                                                                   | /         | Formatted: English (U.S.) |  |
| 330  | spectrum in the spectral range from 245 to 700 min with 5 min resolution; (requation 15),                                                                                                |           |                           |  |
| 351  | $a_{\text{CDOM}}(\lambda) = 10^{(A(\lambda)(\log Chla)^2 + B(\lambda)\log Chla + D(\lambda))},$ (13)                                                                                     |           | Formatted                 |  |
|      |                                                                                                                                                                                          |           | Formatted: English (U.S.) |  |
| 352  | where $A(\lambda)$ [m -2 mg -2 ], $B(\lambda)$ [m -2 mg -1 ] and $\overline{D}(\lambda)$ [m -1 ] are the regression coefficients. | $\langle$ | Formatted: English (U.S.) |  |
| 353  | The spectral distribution of the regression coefficients and determination coefficient                                                                                                   |           | Formatted                 |  |
| 354  | have been are presented oin Figure 4 and their values weare included in Table A in Appendix                                                                                              | $\square$ |                           |  |
| 355  | A. Both regression coefficients $A(\lambda)$ and $B(\lambda)$ showed exhibited a relatively small spectral                                                                               |           |                           |  |
| 356  | variation in the UV and part of the visible spectral range. The biggest changes in regression                                                                                            |           |                           |  |
| 357  | coefficients spectra have been were noted above 580 nm, where a significant increase of the                                                                                              |           |                           |  |
| 358  | in $A(\lambda)$ was to a large extent has been relatively compensated with by a decrease of the in $\beta$                                                                               |           |                           |  |
| 359  | $B(\lambda)$ . Solution Spectral distribution of regression coefficient A, indicates a potential $\beta$                                                                                 |           |                           |  |
| 360  | influence of the phytoplankton pigments absorption on the CDOM absorption spectrum, as its                                                                                               |           |                           |  |
| 361  | maximum, situated around 675 nm, overlaps with the longwave maximum of the chlorophyll                                                                                                   |           |                           |  |
| 362  | a absorption spectrum. This effect is visible apparent only at longer wavelengths, because the                                                                                           |           |                           |  |
| 363  | principale chlorophyll a maximum at 440 nm, is masked by CDOM absorption, especially at                                                                                                  |           |                           |  |
| 364  | in very turbid estuarine and fresh water, where the highest values of CDOM absorption were                                                                                               |           |                           |  |
| 365  | recorded. The free term $D(\lambda)$ spectrum, decreasesing monotonically with increased                                                                                                 |           |                           |  |
| 366  | wavelength, resembles theat of the logtransformed CDOM absorption coefficient spectrum                                                                                                   |           |                           |  |
| 367  | corresponding to the average CDOM absorption spectrum at a chlorophyll a concentration of                                                                                         |           |                           |  |
| 368  | 1 mg m -3 -, as shown oin Figure 4-c. The spectral distribution of the determination coefficient                                                                              |           |                           |  |
| 369  | values R 2 ( , presented on Figure 4-d) , shows demonstrated that, the model based on the                                                                              |           |                           |  |
| 370  | dependency between the CDOM absorption coefficient and the chlorophyll $p$ concentration,                                                                                                |           |                           |  |
| 371  | explained more than 80% of the variability in $a_{\text{CDOM}}(\lambda)$ in the UV and VIS, and that this                                                                                |           |                           |  |
| 372  | variability was controlled governed by phytoplankton biomass production. The model's                                                                                                     |           |                           |  |
| 373  | performance deteriorated at wavelengths longer than 550 nm.                                                                                                                              |           | Formatted: English (U.S.) |  |
|      | 12                                                                                                                                                                                       |           |                           |  |

| 374 | The results of the model uncertainty analysis result for selected wavelengths have                             | Formatted                 |
|-----|----------------------------------------------------------------------------------------------------------------|---------------------------|
| 375 | been-are summarized in Table 3 and presented-illustrated oin Figure 5. Comparison between                      |                           |
| 376 | estimated vs. and measured $a_{\text{CDOM}}(\lambda)$ values at selected wavelengths (260, 350, 440, 500, 550, |                           |
| 377 | 600 nm) from the range 240 - 700 nm range weare shown on the first six upper panels of                         |                           |
| 378 | Figure 5 (a-f). Histograms of the ratios of between estimated and to measured values at the                    |                           |
| 379 | same wavelengths, weare presented oin the lower six Figure 5 panels of Figure 5 (g-1). The                     |                           |
| 380 | deterioration of model performance with increasing wavelength has been is evident. The                         |                           |
| 381 | overall uncertainty expressed by arithmetic statistics and logarithmic statistics wajes                        |                           |
| 382 | satisfactory up to 500 nm, and but then both systematic and statistical estimation errors                      |                           |
| 383 | increasedd rapidly at longer wavelengths. The arithmetic systematic error has increased from                   |                           |
| 384 | 1.47% at 260 nm to 19.54% at 600 nm, and the arithmetic statistical error has increased from                   |                           |
| 385 | 17.03% at 260 nm <del>,</del> to 79.13% at 600-respectively. Logarithmic uncertainty metrics indicated         |                           |
| 386 | that, the standard error factor estimated for the entire spectral range from 240 to 700 nm of                  |                           |
| 387 | light absorption coefficients varieds from 1.19 to 2.66. This meanst that the statistical                      |                           |
| 388 | logarithmic error variesd from -62% to +165%. The logarithmic systematic errors throughout                     |                           |
| 389 | in-the all-240 - 700 nm range doid not exceed 3%.                                                              | Formatted: English (U.S.) |
|     |                                                                                                                | Formatted                 |
| 390 | 3.3. An empirical model for approximating on of the CDOM light absorption spectrum based                       | ()                        |
| 391 | on the empirical dependence on with the CDOM absorption coefficient value at 400 nm,                    |                           |
| 392 | $a_{\text{CDOM}}(400)$                                                                                         | Formatt